# SineNet: Learning Temporal Dynamics in Time-Dependent Partial Differential Equations

**Xuan Zhang**[1]*, **Jacob Helwig**[1]*, **Yuchao Lin**[1], **Yaochen Xie**[1], **Cong Fu**[1], **Stephan Wojtowytsch**[2] **& Shuiwang Ji**[1]

[1]Department of Computer Science & Engineering, Texas A&M University
[2]Department of Mathematics, University of Pittsburgh

## Abstract

We consider using deep neural networks to solve time-dependent partial differential equations (PDEs), where multi-scale processing is crucial for modeling complex, time-evolving dynamics. While the U-Net architecture with skip connections is commonly used by prior studies to enable multi-scale processing, our analysis shows that the need for features to evolve across layers results in temporally misaligned features in skip connections, which limits the model's performance. To address this limitation, we propose SineNet, consisting of multiple sequentially connected U-shaped network blocks, referred to as waves. In SineNet, high-resolution features are evolved progressively through multiple stages, thereby reducing the amount of misalignment within each stage. We furthermore analyze the role of skip connections in enabling both parallel and sequential processing of multi-scale information. Our method is rigorously tested on multiple PDE datasets, including the Navier-Stokes equations and shallow water equations, showcasing the advantages of our proposed approach over conventional U-Nets with a comparable parameter budget. We further demonstrate that increasing the number of waves in SineNet while maintaining the same number of parameters leads to a monotonically improved performance. The results highlight the effectiveness of SineNet and the potential of our approach in advancing the state-of-the-art in neural PDE solver design. Our code is available as part of AIRS (https://github.com/divelab/AIRS).

## 1 Introduction

Partial differential equations (PDEs) describe physics from the quantum length scale (Schrödinger equation) to the scale of space-time (Einstein field equations) and everywhere in between. Formally, a PDE is an expression in which the rates of change of one or multiple quantities with respect to spatial or temporal variation are balanced. PDEs are popular modeling tools across many scientific and engineering fields to encode force balances, including stresses that are combinations of spatial derivatives, velocity and acceleration that are time derivatives, and external forces acting on a fluid or solid. Due to their widespread use in many fields, solving PDEs using numerical methods has been studied extensively. With advances in deep learning methods, there has been a recent surge of interest in using deep learning methods for solving PDEs (Raissi et al., 2019; Kochkov et al., 2021; Lu et al., 2021; Stachenfeld et al., 2022; Li et al., 2021a; Pfaff et al., 2021; Takamoto et al., 2022; Gupta & Brandstetter, 2023), which is the subject of this work.

Our main focus here is on *fluid dynamics*, where we typically encounter two phenomena: advection and diffusion. Diffusion in isotropic media is modeled using the heat equation $(\partial_t - \Delta)\boldsymbol{u} = 0$, which equates the rate of change in time $\partial_t \boldsymbol{u}$ to the spatial Laplacian of the function $\boldsymbol{u}$, an expression of second-order derivatives. The fact that a first-order derivative in time is compared to a second derivative in space leads to the *parabolic scaling* in which $\sqrt{t}$ and $x$ behave comparably. In many numerical methods such as finite element discretizations, this behavior requires discretizations for which the length-scale in time $\delta t$ is much smaller than the spatial length-scale $\delta x$: $\delta t \ll (\delta x)^2$. To avoid excessive computation time when computing in fine resolution, multi-scale models are attractive

---

*Equal contribution. Correspondence to: Xuan Zhang <xuan.zhang@tamu.edu>, Jacob Helwig <jacob.a.helwig@tamu.edu> and Shuiwang Ji <sji@tamu.edu>.

which accurately resolve both local interfaces and long-range interactions over a larger time interval per computation step. Methods which are not localized in space, *e.g.*, heat kernel convolutions or spectral methods, are able to overcome the parabolic scaling.

Unsurprisingly, the situation becomes much more challenging when advection is considered. While diffusion models how heavily localized quantities spread out to larger regions over time, advection describes the transport of quantities throughout space without spreading out. In the equations of fluid dynamics, the transport term $(\boldsymbol{u} \cdot \nabla)\boldsymbol{u}$ is non-linear in $\boldsymbol{u}$ and famously challenging both analytically and numerically. In particular, any numerical approximation to the solution operator which propagates $\boldsymbol{u}$ forward in time must be non-linear to comply with the non-linearity in the equation.

For its strength in multi-scale processing, the U-Net architecture has recently become a popular choice as an inexpensive surrogate model for classical numerical methods mapping dynamics forward in time (Gupta & Brandstetter, 2023; Takamoto et al., 2022; Wang et al., 2020). However, because the input to the network is earlier in time than the prediction target, latent evolution occurs in the U-Net feature maps. This implies that skip connections between the down and upsampling paths of the U-Net, which are crucial for faithfully restoring high-resolution details to the output, will contain out-of-date information, and poses challenges in settings where advection plays a significant role.

In this work, we propose **SineNet**, an architecture designed to handle the challenges arising from dynamics wherein *both* diffusion and advection interact. SineNet features a multi-stage architecture in which each stage is a U-Net block, referred to as a wave. By distributing the latent evolution across multiple stages, the degree of spatial misalignment between the input and target encountered by each wave is reduced, thereby alleviating the challenges associated with modeling advection. At the same time, the multi-scale processing strategy employed by each wave effectively models multi-scale phenomena arising from diffusion. Based on the proposed SineNet, we further analyze the role of skip connections in enabling multi-scale processing in both parallel and sequential manners. Finally, we demonstrate the importance of selecting an appropriate padding strategy to encode boundary conditions in convolution layers, a consideration we find particularly relevant for periodic boundaries. We conduct empirical evaluations of SineNet across multiple challenging PDE datasets, demonstrating consistent performance improvements over existing baseline methods. We perform an ablation study in which we demonstrate performance monotonically improving with number of waves for a fixed parameter budget. The results highlight the potential of our approach in advancing the state-of-the-art in neural PDE solver design and open new avenues for future research in this field.

## 2 NEURAL APPROXIMATIONS TO PDE SOLUTION OPERATORS

Time-evolving partial differential equations describe the behavior of physical systems using partial derivatives of an unknown multivariate function of space-time $\boldsymbol{u} : \Omega \times \mathbb{R} \to \mathbb{R}^M$, where $\Omega$ is the spatial domain and $\mathbb{R}$ accounts for the temporal dimension. The $M$-dimensional codomain of $\boldsymbol{u}$ consists of scalar and/or vector fields such as density, pressure, or velocity describing the state of the system. Given the solution $\boldsymbol{u}(\cdot, t_0)$ at time $t_0$, our task is to learn the forward operator mapping $\boldsymbol{u}(\cdot, t_0)$ to the solution at a later time $t$ given by $\boldsymbol{u}(\cdot, t)$. In practice, $\boldsymbol{u}$ is a numerical solution to the PDE, that is, discretized in space-time onto a finite grid. As opposed to neural operators (Kovachki et al., 2021), which aim to learn this operator independent of the resolution of the discretization, we instead focus on learning the operator for a fixed, uniform discretization. Concretely, for a $d$-dimensional spatial domain $\Omega$, the discretized solution at time step $t$ is given by $\boldsymbol{u}_t \in \mathbb{R}^{M \times N_1 \times \cdots \times N_d}$ for $t = 1, \ldots, T$, where $T$ is the temporal resolution and $N_1, \ldots, N_d$ are the spatial resolutions along each dimension of $\Omega$ ($d = 2$ in our experiments). Moreover, as is typical in numerical solvers, we condition our models on $h$ historical time steps. Formally, we seek to learn the mapping $\mathcal{M} : \mathbb{R}^{h \times M \times N_1 \times \cdots \times N_d} \to \mathbb{R}^{M \times N_1 \times \cdots \times N_d}$, with $\mathcal{M}(\boldsymbol{u}_{t-h+1}, \ldots, \boldsymbol{u}_t) = \boldsymbol{u}_{t+1}$, for $t = h, \ldots, T - 1$.

## 3 LEARNING TEMPORAL DYNAMICS IN CONTINUUM FIELDS

### 3.1 MULTI-SCALE LEARNING AND U-NETS

In Fourier space, low frequency modes correspond to global information, and high frequency modes to local. Convolutions can be performed in the frequency domain via pointwise multiplication of

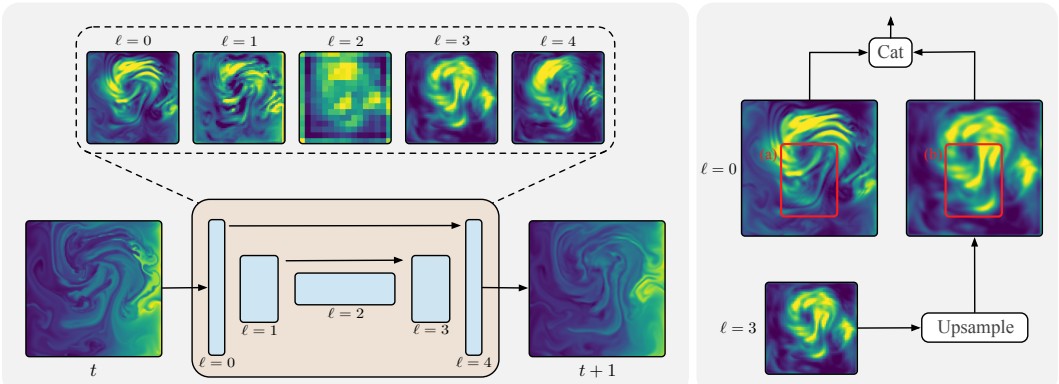

Figure 1: Illustration of the misalignment issue in U-Nets. The left side shows the one-step prediction of a trained U-Net, with each intermediate layer's feature maps averaged over the channel dimension displayed in the top row. On the right side, the misalignment between $\ell = 0$ and $\ell = 3$ is demonstrated. Specifically, the feature map of $\ell = 3$ is upsampled and then concatenated with $\ell = 0$. The time-evolving effect of the preceding U-Net layers results in a misalignment of the corresponding physical features, as indicated by the bounding boxes (a) and (b). This misalignment is particularly problematic for convolutions. Since the kernel is localized, information from misaligned high frequency features, such as those visualized in (a) and (b), cannot be optimally integrated in updating the feature map. Mitigating this misalignment is key for improving the performance of U-Nets.

frequency modes, resulting in parallel processing of multi-scale information (Gupta & Brandstetter, 2023). This approach is taken by Fourier Neural Operators (FNOs) (Li et al., 2021a; Tran et al., 2023; Li et al., 2022; Rahman et al., 2022; Helwig et al., 2023), which directly parameterize their convolution kernel in the frequency domain.

In the spatial domain, U-Nets (Ronneberger et al., 2015) are commonly used to process multi-scale information in a sequential manner. U-Nets extract information on various scales hierarchically by performing convolutions on increasingly downsampled feature maps before invoking a symmetric upsampling path. Skip connections between corresponding downsampling and upsampling layers are established to restore high-resolution information. To map $\{\boldsymbol{u}_{t-h+1}, \ldots, \boldsymbol{u}_t\}$ to $\boldsymbol{u}_{t+1}$, the U-Net encodes inputs to $\boldsymbol{x}_0$ and then processes $\boldsymbol{x}_0$ to a sequence of $L$ feature maps along a downsampling path $\boldsymbol{x}_1, \ldots, \boldsymbol{x}_L$ as

$$\boldsymbol{x}_0 = P(\{\boldsymbol{u}_{t-h+1}, \ldots, \boldsymbol{u}_t\}); \; \boldsymbol{x}_\ell = f_\ell \left(d\left(\boldsymbol{x}_{\ell-1}\right)\right), \; \ell = 1, \ldots, L, \tag{1}$$

where $P$ is an encoder, $d$ is a downsampling function, and $f_\ell$ composes convolutions and non-linearities. Next, the U-Net upsamples to a sequence of feature maps $\boldsymbol{x}_{L+1}, \ldots, \boldsymbol{x}_{2L}$ along the upsampling path and decodes the final feature map $\boldsymbol{x}_{2L}$ to $\boldsymbol{u}_{t+1}$ as

$$\boldsymbol{x}_\ell = g_\ell \left(v\left(\boldsymbol{x}_{\ell-1}\right), \boldsymbol{x}_{2L-\ell}\right), \; \ell = L+1, \ldots, 2L; \; \boldsymbol{u}_{t+1} = Q\left(\boldsymbol{x}_{2L}\right), \tag{2}$$

where $Q$ is the decoder, $v$ is an upsampling function, and $g_\ell$ concatenates the upsampled feature map with the skip connected downsampled feature map before applying convolutions and non-linearities.

## 3.2 Misalignments in learning temporal dynamics

U-Nets capture global context by aggregating localized features with downsampling layers. Using this global context, the output is constructed with upsampling layers, wherein skip connections to the downsampled feature maps play a key role in restoring high-resolution details. While we consider their application as neural solvers, U-Nets were originally developed for image segmentation (Ronneberger et al., 2015). In segmentation tasks, features in the input image and target segmentation mask are spatially aligned. By contrast, in the context of learning temporal dynamics, the spatial localization property no longer holds, as temporal evolution results in *misalignment* between the input and prediction target. Here, misalignment refers to the displacement of local patterns between two feature maps, which occurs between the PDE solutions at two consecutive time steps due to advection. To resolve this misalignment, feature maps undergo latent evolution. However, due to the local

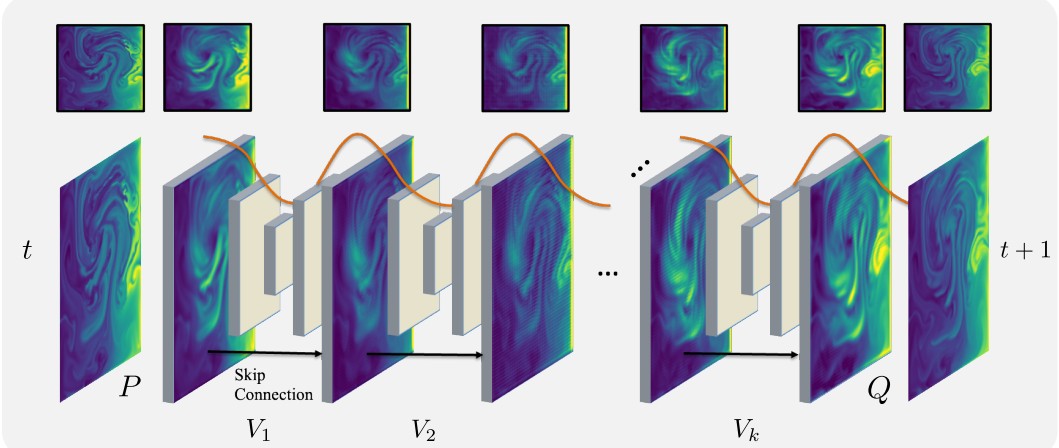

Figure 2: Illustration of the proposed SineNet for learning temporal dynamics in PDEs. Multiple U-Net waves are composed to perform one-step prediction, with the output of each wave averaged over the channel dimension displayed in the top row, demonstrating the time-evolving process from $t$ to $t + 1$. The orange sinusoidal line illustrates propagation between resolutions and is not part of the model architecture. Feature maps in this figure are from SINENET without wave residuals for clarity and with transposed convolutions for upsampling. In Appendix A.2, we visualize the feature maps from the SINENET-8 presented in Section 5.3

processing of convolution layers, feature maps closer to the input (or output) will be more aligned with the input (or output), respectively. This leads to misalignment between the feature maps propagated via skip connections and those obtained from the upsampling path.

To empirically demonstrate this inconsistency, we visualize the feature maps averaged over the channel dimension at each block within a well-trained U-Net in Figure 1. Specifically, we examine the feature map from the skip connection at the highest resolution ($\ell = 0$) and compare it with the corresponding one from the upsampling path ($\ell = 3$). We observe misaligned features between $\ell = 0$ and the upsampled $\ell = 3$, as highlighted by the features contained in the bounding boxes (a) and (b). As kernels are most often local, $e.g.$, $3 \times 3$, following the concatenation, convolution operations will be either partially or fully unable to incorporate the evolved features with the corresponding features in skip connections as a result of the misalignment.

### 3.3 MULTI-STAGE MODELING OF TEMPORAL DYNAMICS WITH SINENET

Despite the inconsistency, skip connections in U-Nets remain crucial to restoring high-frequency details and thereby enable multi-scale computation via downsampling and upsampling. To mitigate the misalignment issue while maintaining the benefits of the U-Net architecture, we construct the SINENET-$K$ architecture by partitioning the latent evolution across $K$ lightweight U-Nets $V_k$, referred to as waves. This approach reduces the latent evolution between consecutive downsampling and upsampling paths, which in turn reduces the degree of misalignment in skip connections. Each wave $V_k$ is responsible for advancing the system's latent state by an interval of $\delta_k \in [0, 1]$, with $\sum_{1 \leq k \leq K} \delta_k = 1$. Specifically, $V_k$'s objective is to evolve the latent solution from time $t + \Delta_{k-1}$, achieved through the temporal evolution by the preceding $k - 1$ waves, to time $t + \Delta_k$, where $\Delta_k$ represents the cumulative sum of all $\delta_j$ up to the $k$-th interval. The overall mapping is then given by

$$\boldsymbol{x}_t = P\left(\{\boldsymbol{u}_{t-h+1}, \ldots, \boldsymbol{u}_t\}\right); \quad \boldsymbol{u}_{t+1} = Q\left(\boldsymbol{x}_{t+1}\right); \tag{3}$$

$$\boldsymbol{x}_{t+\Delta_k} = V_k(\boldsymbol{x}_{t+\Delta_{k-1}}), \; k = 1, \ldots, K, \tag{4}$$

where $\boldsymbol{x}_{t+\Delta_k}$ denotes the latent map of $\boldsymbol{u}_{t+\Delta_k}$ at time $t + \Delta_k$, and $P$ and $Q$ are 3×3 convolution layers to linearly encode and decode the solutions into or from latent maps, respectively. During training, the intermediate sub-steps as determined by $\delta_k$ are implicit and optimized end-to-end with the $K$ waves using pairs of data $\{(\{\boldsymbol{u}_{t-h+1}^j, \ldots, \boldsymbol{u}_t^j\}, \boldsymbol{u}_{t+1}^j)\}_{j,t}$ to minimize the objective

$$\mathbb{E}_{j,t}\left[\mathcal{L}\left(W_K(\{\boldsymbol{u}_{t-h+1}^j, \ldots, \boldsymbol{u}_t^j\}), \boldsymbol{u}_{t+1}^j\right)\right]; \; W_K := Q \circ V_K \circ \cdots \circ V_1 \circ P, \tag{5}$$

where $\mathcal{L}$ is a suitably chosen loss and $j$ indexes the solutions in the training set. It is worth noting that the intervals $\delta_k$ are not necessarily evenly divided. They can be input-dependent and optimized for superior performance during training to enable **adaptable temporal resolution** in the latent evolution. For example, when the temporal evolution involves acceleration, it is generally more effective to use smaller time steps for temporal windows with larger velocities. We demonstrate a connection between this formulation and Neural ODE (Chen et al., 2018) in Appendix F.2.

The proposed approach effectively reduces the time interval each wave is tasked with managing, thereby simplifying the learning task and mitigating the extent of misalignment in skip connections. We empirically demonstrate such effect in Appendix A.1. This strategy not only fosters a more efficient learning process but also significantly enhances the model's ability to handle complex, time-evolving PDEs. Through this approach, we present a robust way to reconcile the essential role of skip connections with the challenges they pose in the context of PDEs.

## 3.4 Wave architecture

Here we discuss the construction of each of the $K$ waves $V_k$ in SineNet mapping the latent solution $\boldsymbol{x}_{t+\Delta_{k-1}}$ to $\boldsymbol{x}_{t+\Delta_k}$ in Equation 4. $V_k$ is implemented as a conventional U-Net $U_k$, as described in Equations 1 and 2, with a wave residual as

$$V_k\left(\boldsymbol{x}_{t+\Delta_{k-1}}\right) = \boldsymbol{x}_{t+\Delta_{k-1}} + U_k(\boldsymbol{x}_{t+\Delta_{k-1}}). \tag{6}$$

Each wave is constructed with a downsampling and upsampling path, both of length $L = 4$, *i.e.*, $\boldsymbol{x}_\ell = f_\ell\left(d\left(\boldsymbol{x}_{\ell-1}\right)\right)$, $\ell \in \{1, 2, 3, 4\}$ for downsampling and $\boldsymbol{x}_\ell = g_\ell\left(v\left(\boldsymbol{x}_{\ell-1}\right), \boldsymbol{x}_{8-\ell}\right)$, $\ell \in \{5, 6, 7, 8\}$ for upsampling. To maintain a light-weight architecture, the downsampling and upsampling functions $d$ and $v$ are chosen as average pooling and bicubic interpolation. $f_\ell^k$ is constructed with a convolution block $c_\ell^k$, that is, two $3 \times 3$ convolutions with layer norm (Ba et al., 2016) and GeLU activation (Hendrycks & Gimpel, 2016), with the first convolution increasing the number of channels in the feature map by a multiplier $m_K$, which is chosen dependent on the number of waves $K$ in the SineNet such that the number of parameters is roughly constant in $K$. $g_\ell^k$ is constructed similarly, except the first convolution *decreases* the number of channels by a factor of $m_K$. Both $f_\ell^k$ and $g_\ell^k$ additionally include a *block residual*, whose role we discuss further in Section 3.5, as

$$f_\ell^k\left(d\left(\boldsymbol{x}\right)\right) = w_\ell^k\left(d\left(\boldsymbol{x}\right)\right) + c_\ell^k\left(d\left(\boldsymbol{x}\right)\right) \tag{7}$$

$$g_\ell^k\left(v\left(\boldsymbol{x}\right), \boldsymbol{y}\right) = w_\ell^k\left(v\left(\boldsymbol{x}\right)\right) + c_\ell^k\left(\mathrm{cat}\left(v\left(\boldsymbol{x}\right), \boldsymbol{y}\right)\right), \tag{8}$$

where $w_\ell^k$ is a point-wise linear transformation acting to increase or decrease the number of channels in the residual connection.

## 3.5 Dual multi-scale processing mechanisms

While in Equation 6, wave residuals serve to improve the optimization of SineNet (He et al., 2016), the block residuals utilized in Equation 7 additionally allow each wave $V_k$ to process multi-scale information following both the sequential and parallel paradigms analyzed by Gupta & Brandstetter (2023). Parallel processing mechanisms, of which the Fourier convolutions utilized by FNOs (Li et al., 2021a) are an example, process features *directly from the input* at various spatial scales independently of one another. In contrast, under the sequential paradigm inherent to U-Nets, features at each scale are extracted from the features extracted at the previous scale. Because this approach does not include a direct path to the input features, processing capacity at a given scale may be partially devoted toward maintaining information from the input for processing at later scales rather than extracting or evolving features.

To improve the efficiency of the sequential paradigm, SineNet utilizes block residuals as described in Equations 7 and 8, which we now show allows both parallel and sequential branches of processing. Consider the input $\boldsymbol{x}_0$, a linear downsampling function $d$ such as average pooling, and convolution blocks $c_1, c_2, c_3$, all of which are composed as the downsampling path of a U-Net, that is, downsampling followed by convolution blocks. Assume for simplicity that the input and output channels for each of the convolution blocks are equal and $w_\ell^k$ is omitted. Then, by including block residuals, the feature map following the first downsampling and convolution block is given by

$$\boldsymbol{x}_1 = d\left(\boldsymbol{x}_0\right) + c_1\left(d\left(\boldsymbol{x}_0\right)\right), \tag{9}$$

and thus, the input to $c_2$ is given by

$$d\left(\boldsymbol{x}_1\right) = d^2\left(\boldsymbol{x}_0\right) + d\left(c_1\left(d\left(\boldsymbol{x}_0\right)\right)\right), \tag{10}$$

where $d^2\left(\boldsymbol{x}_0\right) \coloneqq d\left(d\left(\boldsymbol{x}_0\right)\right)$ represents the direct flow of information from the input in the parallel branch, enabling $c_2$ to process information following both the parallel and sequential paradigms. From the input to $c_3$, given by

$$d\left(\boldsymbol{x}_2\right) = d^3\left(\boldsymbol{x}_0\right) + d\left(d\left(c_1\left(d\left(\boldsymbol{x}_0\right)\right)\right) + c_2\left(d\left(\boldsymbol{x}_1\right)\right)\right), \tag{11}$$

it can be seen that this dual-branch framework is maintained in later layers. Specifically, the input to the $k$-th layer will be comprised of $\boldsymbol{x}_0$ downsampled $k$ times summed with a feature map processed sequentially. However, since addition will entangle the parallel and sequential branch, we propose to disentangle the parallel branch by concatenating it to the input of each convolution block in the downsampling path. Therefore, the downsampling path in SineNet replaces Equation 7 with

$$f_\ell^k\left(d\left(\boldsymbol{x}\right)\right) = w_\ell^k\left(d\left(\boldsymbol{x}\right)\right) + c_\ell^k\left(\operatorname{cat}\left(d\left(\boldsymbol{x}\right), d^k\left(\boldsymbol{x}_0\right)\right)\right), \tag{12}$$

where $d^k\left(\boldsymbol{x}_0\right)$ is the result of the projection layer $P$ downsampled $k$ times. We note that although latent evolution in the sequential branch will result in misalignment with the parallel branch, we empirically observe a performance gain by including the parallel branch, particularly once it has been disentangled. Additionally, this inconsistency between branches is mitigated by the multi-stage processing strategy adopted by SineNet.

### 3.6 ENCODING BOUNDARY CONDITIONS

The boundary conditions of a PDE determine the behavior of the field along the boundary. Our experiments with the incompressible Navier-Stokes equations use a Dirichlet boundary condition on the velocity field such that the velocity on the boundary is zero, and a Neumann boundary condition on the scalar particle concentration such that the spatial derivative along the boundary is zero. These conditions are encoded in feature maps using standard zero padding.

The remaining PDEs we consider have periodic boundary conditions, wherein points on opposite boundaries are identified with one another such that the field "wraps around". For example, for a function $\boldsymbol{u}$ on the unit torus, that is, a periodic function with domain $[0,1]^2$, we have that for $x \in [0,1]$, $\boldsymbol{u}(x,0) = \boldsymbol{u}(x,1)$ and $\boldsymbol{u}(0,x) = \boldsymbol{u}(1,x)$. For such a boundary condition, we found circular padding to be a simple yet crucial component for achieving optimal performance (Dresdner et al., 2023), as otherwise, a great deal of model capacity will be spent towards sharing information between two boundary points that appear spatially distant, but are actually immediately adjacent due to periodicity. As Fourier convolutions implicitly assume periodicity, FNO-type architectures are are ideally suited for such PDEs.

## 4 RELATED WORK

**Neural PDE solvers.** Many recent studies explore solving PDEs using neural networks and are often applied to solve time-dependent PDEs (Poli et al., 2022; Lienen & Günnemann, 2022). Physics-informed neural networks (PINNs) (Raissi et al., 2019; Wang et al., 2021; Li et al., 2021b) and hybrid solvers (Um et al., 2020; Kochkov et al., 2021; Holl et al., 2020b) share similar philosophies with classical solvers, where the former directly minimizes PDE objectives and the latter learns to improve the accuracy of classical solvers. On the other hand, many works focus on purely data-driven learning of mappings between PDE solutions, which is the approach we take here. Wang et al. (2020) and Stachenfeld et al. (2022) apply their convolutional architectures to modeling turbulent flows in 2 spatial dimensions, while Lienen et al. (2023) use diffusion models to simulate 3-dimensional turbulence. Li et al. (2021a) developed the FNO architecture which has been the subject of several follow-up works (Poli et al., 2022; Tran et al., 2023; Helwig et al., 2023). Similar to SineNet, PDE-Refiner (Lippe et al., 2023) solves PDEs via iterative application of a U-Net, however, instead of advancing time with each forward pass, U-Net applications after the first serve to refine the prediction.

**Stacked U-Nets.** There are other studies exploring stacked U-Nets, however, most focus on computer vision tasks such as image segmentation (Xia & Kulis, 2017; Shah et al., 2018; Zhuang, 2018; Fu et al., 2019) and human pose estimation (Newell et al., 2016). There are also researchers exploring

U-Nets and variants therein to model PDE solutions. Chen et al. (2019) apply variations of U-Nets, including stacked U-Nets, to predict the steady state of a fluid flow, although they do not study temporal evolution and furthermore do not consider a stack greater than 2. Raonić et al. (2023) propose to use CNNs to solve PDEs, but they do not have the repeated downsampling and upsampling wave structure. We differ from these works by making the key observation of feature evolution and feature alignment.

## 5 EXPERIMENTS

As fluid dynamics are described by time-evolving PDEs where diffusion and advection play major roles, we perform experiments on multiple fluid dynamics datasets derived from various forms of the Navier-Stokes equation. We begin by describing our experimental setup, datasets and models in Sections 5.1-5.2 before presenting our primary results in Section 5.3. We close with an abalation study in Section 5.4. Furthermore, in Appendix A.1, we conduct an experiment to validate our claim of reduced latent evolution per wave, and demonstrate that this results in reduced misalignment in skip connections in Appendix A.2.

### 5.1 SETUP AND DATASETS

In all experiments, models are trained with data pairs $(\{\boldsymbol{u}_{t-h+1}, \ldots, \boldsymbol{u}_t\}, \boldsymbol{u}_{t+1})$ where the inputs are the fields at current and historical steps and the target is the field at the next time step. During validation and testing, models perform an autoregressive rollout for several time steps into the future. Scaled $L_2$ loss (Gupta & Brandstetter, 2023; Tran et al., 2023; Li et al., 2021a) is used as training loss and evaluation metric, which we describe in Appendix D.2. We report both the one-step and rollout test errors.

We consider three datasets in our experiments. Each dataset consists of numerical solutions for a given time-evolving PDE in two spatial dimensions with randomly sampled initial conditions. Detailed dataset descriptions can be found in Appendix B.2.

**Incompressible Navier-Stokes (INS)**. The incompressible Navier-Stokes equations model the flow of a fluid wherein the density is assumed to be independent of the pressure, but may not be constant due to properties of the fluid such as salinity or temperature (Vreugdenhil, 1994). The equations are given by

$$\frac{\partial \boldsymbol{v}}{\partial t} = -\boldsymbol{v} \cdot \nabla \boldsymbol{v} + \mu \nabla^2 \boldsymbol{v} - \nabla p + \boldsymbol{f}, \ \nabla \cdot \boldsymbol{v} = 0, \tag{13}$$

where $\boldsymbol{v}$ is velocity, $p$ is internal pressure, and $\boldsymbol{f}$ is an external force. We use the dataset from Gupta & Brandstetter (2023), simulated with a numerical solver from the $\Phi_{\text{Flow}}$ package (Holl et al., 2020a). In Appendix F.1, we present results on the conditional version of this dataset considered by Gupta & Brandstetter (2023), where the task is to generalize over different time step sizes and forcing terms.

**Compressible Navier-Stokes (CNS)**. Compressibility is generally a consideration most relevant for fast-moving fluids (Anderson, 2017). We generate our CNS dataset using the numerical solver from Takamoto et al. (2022). The dynamics in this data are more turbulent than those in the INS data, with the viscosity as $1 \times 10^{-8}$ and the initial Mach number, which quantifies the ratio of the flow velocity to the speed of sound in the fluid (Anderson, 2017), as 0.1.

**Shallow Water Equations (SWE)**. The shallow water equations are derived by depth-integrating the incompressible Navier-Stokes equations and find applications in modeling atmospheric flows (Vreugdenhil, 1994). We use the dataset from Gupta & Brandstetter (2023) for modeling the velocity and pressure fields for global atmospheric winds with a periodic boundary condition.

### 5.2 MODELS

Here, we overview the models used in our experiments, and provide further details in Appendix D.

**SINENET-8/16.** SineNet with 8 or 16 waves and 64 channels at the highest resolution. Multiplier $m_k = 1.425/1.2435$ is used, resulting in channels along the downsampling path of each wave being arranged as $(64, 91, 129, 185, 263)/(64, 79, 98, 123, 153)$.

**F-FNO.** Fourier Neural Operators (Li et al., 2021a) process multi-scale features by performing convolutions in the frequency domain (Gupta & Brandstetter, 2023), and were originally developed for PDE modeling. As one of the primary components of SineNet is depth, we compare to a state-of-the-art FNO variant, the Factorized FNO (F-FNO) (Tran et al., 2023), optimized specifically for depth. In each of the 24 layers, the number of Fourier modes used is 32 and the number of channels is 96.

**DIL-RESNET.** As opposed to downsampling and upsampling, the dilated ResNet (DIL-RESNET) proposed by Stachenfeld et al. (2022) is a neural PDE solver that processes multi-scale features by composing blocks of convolution layers with sequentially increasing dilation rates followed by sequentially decreasing dilation rates (Zhang et al., 2023).

**U-NET-128 and U-NET-MOD.** U-NET-128 is equivalent to SINENET-1 with the multiplier $m_K = 2$ and 128 channels at the highest resolution, with channels along the downsampling path arranged as $(128, 256, 512, 1024, 2048)$. U-NET-MOD is a modern version of the U-Net architecture which parameterizes downsampling using strided convolutions and upsampling using transposed convolutions. Additionally, it doubles the number of convolutions and skip connections per resolution.

For convolution layers in SINENET, DIL-RESNET, U-NET-128 and U-NET-MOD, zero padding is used on the INS dataset and circular padding is used on CNS and SWE.

## 5.3 RESULTS

We present results for INS, CNS, and SWE in Table 1. On all datasets, SINENET-8 has the lowest 1-step and rollout errors. Out of the remaining baselines, F-FNO has the strongest performance in terms of rollout error on CNS and SWE, while DIL-RESNET has the best baseline rollout error on INS. We visualize SINENET-8 predictions on each dataset in Appendix G.

Table 1: Summary of rollout test error and one-step test error. The best performance is shown in bold and the second best is underlined.

| | | INS | | CNS | | SWE | |
| --- | --- | --- | --- | --- | --- | --- | --- |
| METHOD | # PAR. (M) | 1-STEP (%) | ROLLOUT (%) | 1-STEP (%) | ROLLOUT (%) | 1-STEP (%) | ROLLOUT (%) |
| SINENET-8 | 35.5 | 1.66 | 19.25 | 0.93 | 2.10 | 1.02 | 1.78 |
| SINENET-16 | 35.4 | **1.46** | **17.63** | **0.87** | **2.04** | **0.92** | **1.64** |
| F-FNO | 30.1 | 2.41 | 22.58 | 1.46 | 2.91 | 1.22 | 2.46 |
| DIL-RESNET | 16.7 | 1.72 | 19.29 | 1.17 | 3.76 | 2.23 | 4.12 |
| U-NET-128 | 135.1 | 2.69 | 24.94 | 1.62 | 3.05 | 1.63 | 3.38 |
| U-NET-MOD | 144.3 | 2.43 | 23.65 | 1.32 | 3.34 | 1.60 | 3.02 |

## 5.4 ABLATION STUDY

Table 2: Ablation study results. The best performance is shown in bold.

| | | INS | | CNS | | SWE | |
| --- | --- | --- | --- | --- | --- | --- | --- |
| METHOD | # PAR. (M) | 1-STEP (%) | ROLLOUT (%) | 1-STEP (%) | ROLLOUT (%) | 1-STEP (%) | ROLLOUT (%) |
| SINENET-8 | 35.5 | **1.66** | **19.25** | **0.93** | **2.10** | **1.02** | **1.78** |
| SINENET8-ENTANGLED | 32.8 | 1.69 | 19.46 | 1.01 | 2.39 | 1.14 | 1.97 |
| DEEPER U-NET-8 | 28.6 | 1.74 | 19.58 | 1.21 | 2.76 | 1.39 | 2.67 |

We construct several models to ablate various components of SineNet and present results for these models in Table 2.

**DEEPER U-NET-8.** Conventional U-Net, but increases the number of convolutions per resolution along the down and upsampling paths by a factor of 8 relative to SINENET-8, resulting in an architecture with a similar number of layers to SINENET-8, as visualized in Figure 6 in Appendix D. This serves to ablate SineNet's multi-stage processing strategy as a means of resolving the misalignment issue, and furthermore validate that observed improvements of SineNet are not a mere result of increased depth. In Table 2, we see that on all datasets, SINENET-8 outperforms DEEPER U-NET-8. The largest improvements are on the SWE data, where the time interval between consecutive solutions is 48 hours. While the large timestep results in substantial misalignment in the DEEPER U-NET skip connections, the multi-stage strategy of SINENET handles it effectively.

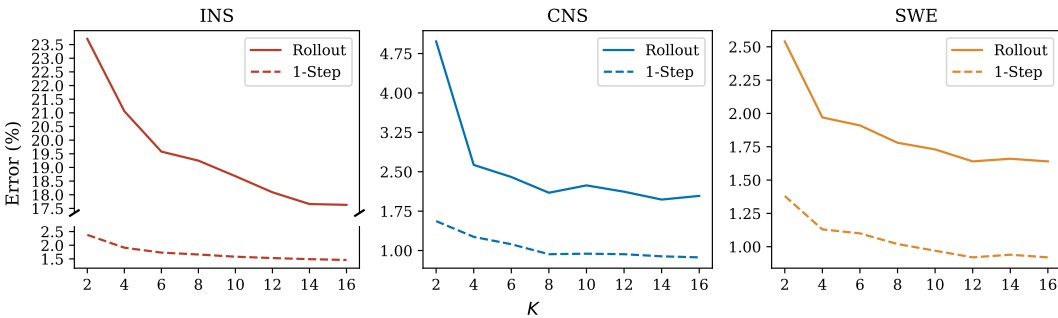

Figure 3: Number of waves $K$ versus test error. Solid line is rollout error, dashed line is 1-step error. Numerical results are presented in Appendix F.8.

**SINENET-8-ENTANGLED.** SINENET-8, but the parallel and sequential processing branches are entangled, *i.e.*, the downsampling path is constructed as in Equation 7 as opposed to the disentangled downsampling in Equation 11. In Table 2, we observe that disentangling the parallel branch results in consistent performance gains.

**SINENET-$K$.** SINENET with $K$ waves, and the channel multiplier $m_K$ chosen such that the number of parameters is roughly constant across all models, which we discuss further in Appendix D.3. In Figure 3, we present results for $K = 2, 4, 6, 8, 10, 12, 14, 16$ and find that on all 3 datasets, errors monotonically improve with $K$, although improvements appear to plateau around $K = 16$. This result is consistent with our assumption of the latent evolution of features, as a greater number of waves leads to a smaller amount of evolution managed by each wave, thereby leading to improved modeling accuracy through reduced misalignment. In Appendix C, we present the inference time and space requirements for SINENET-$K$.

**Effect of circular padding.** In Table 3, we replace circular padding with zero padding in SINENET-8 on SWE. The periodic boundaries result in boundary points opposite each other being spatially close, however, without appropriate padding to encode this into feature maps, the ability to model advection across boundaries is severely limited.

Table 3: Comparison between zero and circular padding on SWE with SINENET-8.

| ZERO PADDING | | CIRCULAR PADDING | |
|---|---|---|---|
| 1-STEP (%) | ROLLOUT (%) | 1-STEP (%) | ROLLOUT (%) |
| 1.50 | 4.19 | **1.02** | **1.78** |

## 6 DISCUSSION

We discuss potential limitations in Appendix E, including out-of-distribution (OOD) generalization, applicability to PDEs with less temporal evolution, computational cost, and handling irregular spatiotemporal grids. Additionally, while SineNet is a CNN and therefore cannot directly generalize to discretizations differing from that in the training set, inference at increased spatial resolution can be achieved through dilated network operations or interpolations without any re-training, which we discuss and experiment with further in Appendix F.3.

## 7 CONCLUSION

In this work, we have presented SineNet, a neural PDE solver designed to evolve temporal dynamics arising in time-dependent PDEs. We have identified and addressed a key challenge in temporal dynamics modeling, the misalignment issue, which results in a performance drop for conventional U-Nets. By reframing the U-Net architecture into multiple waves, SineNet mitigates this misalignment, leading to consistent improvements in solution quality over a range of challenging PDEs. Further, we analyze the role of skip connections in enabling both parallel and sequential processing of multi-scale information. Additionally, we demonstrate that increasing the number of waves, while keeping the number of parameters constant, consistently improves the performance of SineNet. Our empirical evaluation across multiple challenging PDE datasets highlights the effectiveness of SineNet and its superiority over existing baselines.

## ACKNOWLEDGMENTS

This work was supported in part by National Science Foundation grants IIS-2243850 and IIS-2006861.

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

# APPENDIX

## A MISALIGNMENT ANALYSIS

### A.1 SLOWING DOWN EVOLUTION IN LATENT SPACE

We now conduct an empirical analysis to validate our hypothesis concerning the manageable inconsistency due to reduced misalignment within each wave. Specifically, we demonstrate the progressive evolution of the system's state over the waves by injecting a perturbation into the model input and monitoring the ensuing response in the latent maps at each wave. The response to this perturbation serves as an indicator of the influence of the perturbation at each time step, thereby enabling us to track the evolution of the system. To quantify this response, we first carry out a forward pass of the input $\boldsymbol{u}_t$ through our trained model and capture the $\ell$-th feature map, denoted as $\boldsymbol{x}_{t+\Delta\ell}$. We then introduce a localized perturbation to $\boldsymbol{u}_t$ in the form of random noise concentrated in a small central region and re-record the $\ell$-th feature map, denoted as $\tilde{\boldsymbol{x}}_{t+\Delta\ell}$. The absolute difference $\boldsymbol{a}_{t+\Delta\ell} \coloneqq |\tilde{\boldsymbol{x}}_{t+\Delta\ell} - \boldsymbol{x}_{t+\Delta\ell}|$ is computed and visualized to represent the response. We add such noise to samples from the test set as it allows us to see the feature propagation from an excitation with a small spatial extent. Otherwise, if we only provide the per-

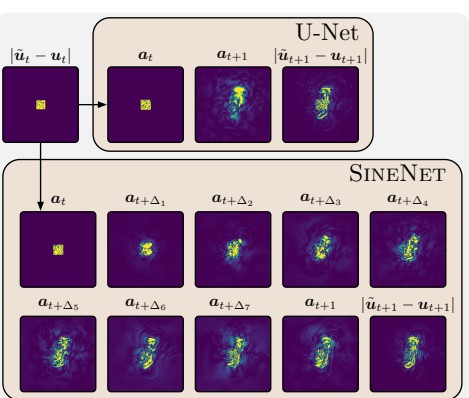

Figure 4: Feature map responses to noise injection for trained U-NET-128 and SINENET-8. As opposed to the U-NET, in SINENET, the perturbation influence propagates gradually from the first to the last feature map, demonstrating the reduced latent evolution managed by each wave.

turbation as input to the network, it will become an out-of-distribution problem for the network since there are no similar samples in the training set. However, if we inject a small amount of noise to a sample in the test set, the resulting input remains close to the original test sample, and therefore is still in-distribution and can be used to analyze the behavior of the trained model.

Figure 4 showcases the evolution of this response across the latent maps $\boldsymbol{a}_{t+\Delta_0}, \ldots, \boldsymbol{a}_{t+\Delta_K}$ in a well-trained SINENET-8 model. For comparison, we also include the response from a well-trained U-NET-128, using the output of the first projection layer and the output of the last upsampling block. In both the U-NET and SINENET models, the perturbation influence expands spatially from the input to the output. However, for SINENET-8, the perturbation influence propagates incrementally from the first to the last feature map, indicating that the spatial evolution within each wave is reduced. It's important to note that this progressive propagation is not merely a consequence of limited receptive fields, as the receptive field of each wave is substantially larger than the area influenced by the perturbation. Furthermore, the receptive field for each wave of SINENET is equal to that of the U-NET.

### A.2 FEATURE MAP VISUALIZATION

We visualize feature maps for each wave of SINENET-8 in Figure 5 for a randomly selected example from the INS data. Specifically, for each wave, we show the highest-resolution skip connection from the downsampling path alongside the result of the upsampling path to which the skip-connected feature map is concatenated, as due to latent evolution, these two feature maps will have the greatest misalignment out of all of the skip connections in a given wave. Visualization is done by averaging over the channel dimension of each feature map. To increase the overall contrast, we clip the 1st and 99th quantiles. As can be seen relative to the feature maps from the U-Net in Figure 1, the misalignment in skip connections is substantially mitigated thanks to the multi-stage feature evolution in SineNet.

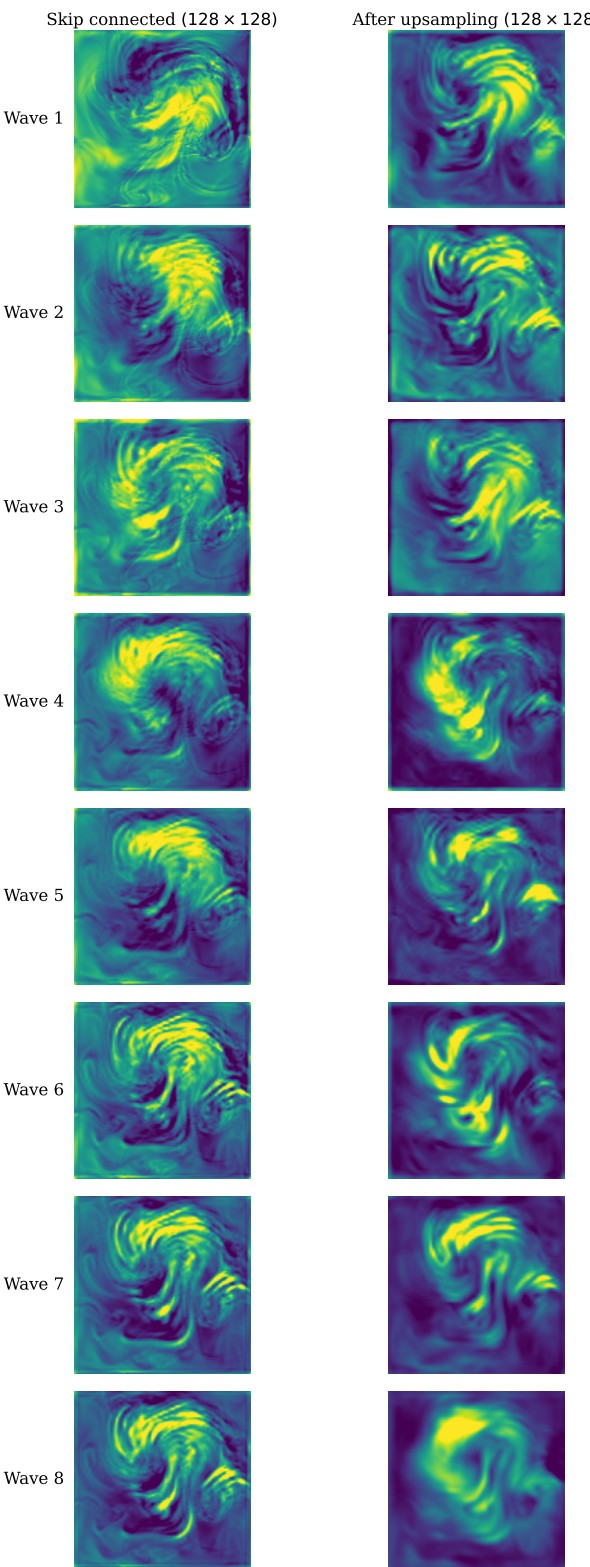

Figure 5: Visualization of SINENET-8 feature maps averaged over the channel dimension for INS data. Skip connections from the highest resolution in the downsampling path of each wave are visualized on the **left**, which are concatenated with the upsampled feature maps visualized on the **right**. Compared to U-Net (Figure 1 in main text), the degree of misalignment in skip connections is far less severe, as feature maps progressively evolve more gradually across waves.

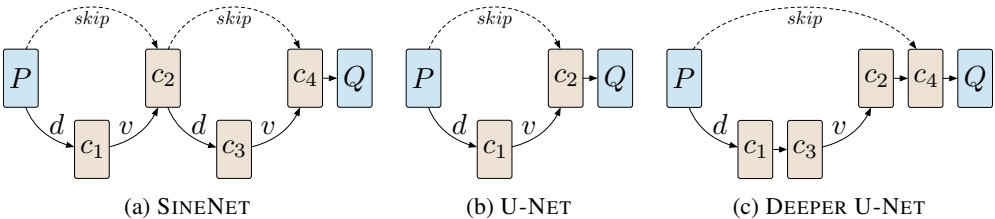

Figure 6: Comparison of SINENET, U-NET and DEEPER U-NET architectures.

## B    DATASET DETAILS

### B.1    THE NAVIER-STOKES EQUATION

The Navier-Stokes equations (Constantin & Foias, 2020) are the fundamental model which encodes the conservation of mass and momentum in fluid flows in manifold applications ranging from weather research to aerospace engineering and stellar magneto-hydrodynamics. It comprises a system of non-linear partial differential equations for the density $\rho$, velocity field $\boldsymbol{u}$, and pressure $p$ of a fluid (compressible case) or velocity $\boldsymbol{u}$ and pressure $p$ (incompressible case). The Navier-Stokes equations are the dominant model of fluid dynamics from which other equations such as the Euler equation or the shallow water equations are derived in limiting regimes (zero viscosity, shallow water). Due to its foundational role, it is the subject of an unsolved Millennium Problem of the Clay Mathematics Institute and has attracted attention from mathematical analysts as well as applied scientists. Challenges abound, especially for fast-moving turbulent flows, and despite recent breakthroughs (Albritton et al., 2022), even the existence and uniqueness of solutions is not fully understood.

### B.2    DETAILED SETUP

**Incompressible Navier-Stokes (INS).** Each trajectory is 14 time steps spaced 1.5 seconds apart and models the evolution of a vector velocity field with a Dirichlet boundary condition and a scalar field with a Neumann boundary conditon representing the concentration of particles advected by the velocity field. Trajectories are spatially discretized onto a $128 \times 128$ grid. The data follow a train/valid/test split of 5,200/1,300/1,300, and the length of the time history $h$ is 4 steps such that trajectories are unrolled for 10 steps during evaluation. The viscosity $\mu$ controlling the level of turbulence in the flow is 0.01.

**Compressible Navier-Stokes (CNS).** Each trajectory has 21 time steps and consists of a scalar pressure field, a scalar density field, and a vector velocity field, each with periodic boundary conditions. Trajectories are generated on a $512 \times 512$ spatial grid and downsampled to $128 \times 128$. The dataset is split as 5,400/1,300/1,300 and, following Takamoto et al. (2022), models use time history $h = 10$ such that trajectories are unrolled for 11 steps during evaluation.

**Shallow Water Equations (SWE).** The trajectories are generated using the numerical solver from Klöwer et al. (2022) on a $96 \times 192$ spatial discretization. Each trajectory consists of 11 time steps, with 48 hours between each step. The data are split as 5,600/1,400/1,400, and $h = 2$ historical time steps are used such that trajectories are unrolled for 9 steps during evaluation.

## C    TIME AND SPACE COMPLEXITY

In Table 4, we analyze the inference time, training time, and GPU footprint for all models on a batch of size 32 randomly selected from the INS dataset. The times presented here are an average over 1,000 batches on a single 80GB A100 GPU.

Table 4: Analysis of inference and training time, as well as memory required for all models on a batch of size 32 randomly selected from the INS dataset. A batch size of 16 is used for SINENET-NEURAL-ODE in the Forward+Backward complexity test as it cannot fit into the GPU memory with batch size 32.

| METHOD | # PAR. (M) | FORWARD | | FORWARD+BACKWARD | |
| | | TIME (S) | MEMORY (GB) | TIME (S) | MEMORY (GB) |
|---|---|---|---|---|---|
| SINENET-2 | 35.5 | 0.122 | 4.19 | 0.398 | 10.63 |
| SINENET-4 | 35.5 | 0.180 | 3.86 | 0.596 | 15.11 |
| SINENET-6 | 35.5 | 0.225 | 3.74 | 0.729 | 19.18 |
| SINENET-8 | 35.5 | 0.286 | 3.54 | 0.917 | 23.31 |
| SINENET-10 | 35.5 | 0.328 | 3.52 | 1.043 | 27.21 |
| SINENET-12 | 35.5 | 0.368 | 3.87 | 1.157 | 30.90 |
| SINENET-14 | 35.4 | 0.410 | 3.70 | 1.286 | 34.61 |
| SINENET-16 | 35.5 | 0.447 | 3.65 | 1.400 | 38.21 |
| SINENET-NEURAL-ODE | 12.7 | 1.822 | 8.60 | 2.977* | 49.53* |
| F-FNO | 30.1 | 0.399 | 3.91 | 1.175 | 41.05 |
| DIL-RESNET-128 | 4.2 | 0.182 | 3.83 | 0.387 | 28.77 |
| DIL-RESNET-256 | 16.7 | 0.469 | 8.37 | 1.044 | 56.93 |
| U-NET-128 | 135.1 | 0.169 | 7.58 | 0.532 | 13.24 |
| U-NET-MOD | 144.3 | 0.097 | 4.71 | 0.236 | 14.12 |

# D  IMPLEMENTATION DETAILS

## D.1  TRAINING

Our code is implemented in PyTorch (Paszke et al., 2019). Models are trained and evaluated on 2 NVIDIA A100 80GB GPUs. All models are optimized for 50 epochs with batch size 32 and the model with the best validation rollout results is used for testing. Following Gupta & Brandstetter (2023), each epoch consists of $T$ iterations over the data. For each trajectory, a start time $t$ is randomly sampled from $t = h, \ldots, T - 1$. Performing $T$ cycles per epoch ensures that each possible 1-step input-target pair $(\{u_{t-h+1}, \ldots, u_t\}, u_{t+1})$ is sampled more than once in expectation for each trajectory. We calculate statistics from the training data along the field dimension and normalize model inputs and targets to have 0 mean and unit variance. During rollouts, we apply the inverse normalization.

All models are optimized with the AdamW optimizer (Kingma & Ba, 2015; Loshchilov & Hutter, 2019), using an initial learning rate of $\eta_{\text{init}} = 2 \times 10^{-4}$, except for the F-FNO on SWE and CNS, where we found a larger learning rate to improve performance (see Table 6). The learning rate was warmed up linearly for 5 epochs from $\eta_{\text{min}} = 1 \times 10^{-7}$ to $\eta_{\text{init}}$ before being decayed for the remaining 45 epochs using a cosine scheduler.

## D.2  LOSS

We use the Scaled-$L_2$ loss for both training and evaluation. We observe that the magnitude can vary significantly across different fields. Therefore, we compute the Scaled-$L_2$ loss separately for different fields to account for differences in magnitude and then calculate the average across all fields. Given a prediction $\hat{u}_t$ and a target $u_t$ at time step $t$, composed of $M$ fields, the 1-step loss is computed as

$$\mathcal{L}^{\text{1-step}}(\hat{u}_t, u_t) = \frac{1}{M} \sum_{k=1}^{M} \frac{\|\hat{u}_t^k - u_t^k\|_2}{\|u_t^k\|_2}, \tag{14}$$

where $u^k$ denotes the $k$-th field of $u$ and the norm is taken over all spatial dimensions. Note that we consider each scalar component as a separate field. For example, the velocities along $x$ direction and $y$ direction are considered as two different fields and are normalized independently. For validation and test, the rollout loss is computed as the average 1-step loss over rollout time steps

$$\mathcal{L}^{\text{rollout}} = \frac{1}{T - h} \sum_{t=h+1}^{T} \mathcal{L}^{\text{1-step}}(\hat{u}_t, u_t), \tag{15}$$

where $T$ is the number of total time steps and $h$ is the number of conditioning historical time steps.

### D.3 Selection of multiplier hyperparameter

Table 5: Multiplier hyperparameter and number of channels in each wave for SINENET-$K$ across various choices of $K$. The multiplier determines how the number of channels is upscaled along the downsampling path and downscaled along the upsampling path. We adjust this multiplier to manage the number of parameters and computational cost of SineNet. We also list the number of channels produced by each of the 4 blocks from the beginning to the end of the downsampling path. The input number of channels to each wave in all models is $64$.

| Method | # Par. (M) | Multiplier | # of Channels |
|---|---|---|---|
| SINENET-2 | 35.5 | 1.8075 | $(115, 209, 377, 683)$ |
| SINENET-4 | 35.5 | 1.6110 | $(103, 166, 267, 431)$ |
| SINENET-6 | 35.5 | 1.5000 | $(96, 144, 216, 324)$ |
| SINENET-8 | 35.5 | 1.4250 | $(91, 129, 185, 263)$ |
| SINENET-10 | 35.5 | 1.3660 | $(87, 119, 163, 222)$ |
| SINENET-12 | 35.5 | 1.3190 | $(84, 111, 146, 193)$ |
| SINENET-14 | 35.4 | 1.2790 | $(81, 104, 133, 171)$ |
| SINENET-16 | 35.5 | 1.2435 | $(79, 98, 123, 153)$ |

We now discuss the choice of the number of channels in the feature maps along the down and upsampling paths. The number of channels in the feature map output by the $\ell$-th downsampling and upsampling blocks, respectively, is given by

$$z_\ell = \left\lfloor m_K^\ell z_0 \right\rfloor, \qquad z_\ell = \left\lfloor m_K^{4-\ell} z_0 \right\rfloor, \ell = 1, \ldots, L \tag{16}$$

where $z_0$ is the number of channels following the projection by the encoder $P$ and there are $L$ downsampling and upsampling blocks. While conventional U-Nets use multiplier $m_K = 2$, we manage the number of parameters and complexity of our SINENET-$K$ architectures by selecting $m_K$ such that the number of parameters is roughly constant in $K$. In Table 5, we present the multiplier $m_K$ and number of channels along the downsampling path for the SINENET-$K$ architecture with varying number of waves $K$. All architectures presented here use $z_0 = 64$.

### D.4 Baseline Tuning

Here we present hyperparameter tuning results for baseline methods on the SWE dataset. As discussed in Appendix D.1, we found that for F-FNO, using a larger learning rate improved results. Additionally, Tran et al. (2023) found that in their setting, sharing weights between Fourier convolution layers improved F-FNO performance. However, we find not sharing to give better results. We additionally found that increasing the number of channels in DIL-RESNET improved performance. We present validation errors for these experiments in Table 6.

Table 6: Best validation results for different baseline hyperparameter settings on SWE. We evaluate F-FNO with different learning rates, as well as with and without weight sharing. We additionally evaluate the effect of number of channels on DIL-RESNET.

| Method | # Par. (M) | 1-Step Valid (%) | Rollout Valid (%) |
|---|---|---|---|
| FFNO, $\eta_{\text{init}} = 1 \times 10^{-3}$ | 30.1 | **1.21** | **2.44** |
| FFNO, $\eta_{\text{init}} = 2 \times 10^{-4}$ | 30.1 | 1.88 | 3.72 |
| FFNO-SHARED, $\eta_{\text{init}} = 1 \times 10^{-3}$ | 3.0 | 1.77 | 3.44 |
| DIL-RESNET-128 | 4.2 | 3.40 | 6.60 |
| DIL-RESNET-256 | 16.7 | **2.21** | **4.09** |

# E  DISCUSSION ON LIMITATIONS

**Out-of-distribution (OOD) generalization.** Unlike classical solvers which are based on intrinsic mathematical model of PDEs and can be applied to almost any initial and boundary conditions, data-driven surrogate models are only guaranteed to generalize well to scenarios which are close to the training distribution. For example, performance drops substantially when simulating the same dynamics but on a larger domain than observed during training (Stachenfeld et al., 2022). It is possible to improve OOD robustness by combining surrogate models with classical solvers (Kochkov et al., 2021). However, pure surrogate models remain attractive due to flexibility in model design and fast inference. It remains to be seen how training methods can be adapted to improve OOD robustness, with initial works in this direction focusing on meta-learning approaches (Wang et al., 2022; Mouli et al., 2023; Kirchmeyer et al., 2022). In Appendix F.5, we found noise injection during training (Sanchez-Gonzalez et al., 2020; Stachenfeld et al., 2022) improved robustness to difficult intial conditions and generalization to rollouts $10\times$ longer than training rollouts.

**PDEs with reduced temporal evolution.** SineNet was developed specifically for time-evolving PDEs wherein the input fields are spatially misaligned with respect to the target fields. Therefore, further experimentation is needed to determine the benefits of SineNet in learning dynamics where time evolution does not play as large of a role, such as the steady-state Darcy flow equations considered by Li et al. (2021a).

**Computational cost.** As shown in Appendix C, using more waves increases the inference time and training memory due to the added depth. Nevertheless, as we did in the experiments, channel multipliers can be adjusted to control the computational cost.

**Irregular spatial grids and time intervals.** For modeling dynamics on irregular discretizations, graph neural networks have been used (Brandstetter et al., 2022; Pfaff et al., 2021; Wu et al., 2022). As with all CNNs, SineNet can only handle inputs on rectangular grids with uniformly spaced mesh points. However, it is possible to extend SineNet to irregular meshes with graph neural networks (Gao & Ji, 2019; Li et al., 2020), where the key idea of combining multi-resolution processing and multi-stage processing remains valid. Furthermore, because SineNet is trained for autoregressive prediction, it can only advance time by a fixed-size timestep $\Delta_t$, and thus, cannot predict the solution at time points that are not multiples of $\Delta_t$. However, as we show in Appendix F.1, SineNet can be trained to generalize over $\Delta_t$.

**Mixed and non-null boundaries.** As discussed in Section 3.6, the Dirichlet boundary conditions we consider for the particle concentration field can be effectively encoded via zero padding. Although the Neumann boundary conditions on the velocity field would ideally be encoded in feature maps using reflection padding to represent the null spatial derivative, encoding both the Neumann boundary condition and the Dirichlet boundary condition is not straightforward. This is because it is unclear which feature maps correspond to the particle concentration, and which correspond to the velocity field. To further complicate matters, feature maps earlier in the architecture may correspond to both. Although here we choose to only use zero-padding for simplicity, future work should explore principled approaches to encoding mixed boundary conditions for convolutional architectures, with Horie & Mitsume (2022) having initiated this line of work for graph neural networks.

Furthermore, while we consider null Dirichlet and null Neumann boundaries here, non-null conditions are also of interest. For non-null Dirichlet conditions, the value of the field on the boundary is known and can be encoded simply by padding with the known value. In the case of the non-null Neumann condition, one approach could be to pad with a finite difference-type approximation using the feature map's boundary value and the boundary derivative given by the condition. Further work is needed to validate the effectiveness of both approaches.

# F  EXTENDED RESULTS

## F.1  CONDITIONAL INS

In the conditional task considered by Gupta & Brandstetter (2023), models are trained to generalize over variable-sized time steps and variable forcing terms. Specifically, in Equation 13, $\boldsymbol{f}$ is an external force acting along the $x$ and $y$ axes as $\boldsymbol{f} = (0, \tilde{f})^\top \in \mathbb{R}^2$. In the conditional task, the $y$-component

Table 7: Summary of rollout test error and one-step test error for various time step sizes on Conditional INS.

| METHOD | # PAR. (M) | 1-STEP (%) | | | | | ROLLOUT (%) |
| --- | --- | --- | --- | --- | --- | --- | --- |
| | | $\Delta_t = 0.375$ | $\Delta_t = 0.75$ | $\Delta_t = 1.5$ | $\Delta_t = 3.0$ | $\Delta_t = 6.0$ | |
| SINENET-8-ADD | 38.0 | 2.29 | 2.82 | 3.82 | 6.84 | 19.79 | 5.27 |
| SINENET-8-ADAGN | 40.3 | **2.05** | **2.59** | **3.63** | **6.64** | **19.21** | **4.60** |
| U-NET-MOD-ADD | 146.6 | 2.46 | 3.23 | 4.94 | 9.50 | 24.69 | 5.85 |
| U-NET-MOD-ADAGN | 148.8 | 2.16 | 2.91 | 4.46 | 8.86 | 24.94 | 5.00 |

of the forcing term varies between trajectories as $f \in [0.2, 0.5]$. Furthermore, the time step size is reduced by a factor of 4 relative to the INS data from 1.5 seconds to 0.375 seconds such that the number of time steps increases from 14 to 56. The one-step training objective then becomes the prediction of $\boldsymbol{u}_{t+\Delta_t}$ given $\boldsymbol{u}_t$, where the time step $\Delta_t$ takes values in $\{0.375k : k = 1, \dots, 55\}$.

To model the variable forcing term and time step size, network inputs are $(\boldsymbol{u}_t, \kappa)$, where $\kappa$ is the time-step and buoyancy $\kappa := (f, \Delta_t)$. Gupta & Brandstetter (2023) condition models on $\kappa$ by learning an embedding $\epsilon$ of $\kappa$ obtained by applying sinusoidal embeddings (Vaswani et al., 2017) and an MLP to $f$ and $\Delta_t$. In each convolution block, $\epsilon$ is linearly projected and added along the channel dimension of feature maps. Gupta & Brandstetter (2023) additionally consider a scale-shift approach, where a linear projection of $\epsilon$ is multiplied element-wise along the channel dimension prior to addition along the channel dimension with a second linear projection of $\epsilon$ (Perez et al., 2018). Gupta & Brandstetter (2023) refer to this approach as *adaptive group normalization*, since the scale and shift are applied directly following normalization (Nichol & Dhariwal, 2021). Since SineNet does not use pre-activation as in the U-NET-MOD considered by Gupta & Brandstetter (2023), wherein normalization and activation functions are applied prior to convolution layers, the scale and shift in SineNet are instead applied following the activation function. Nonetheless, we still refer to the scale-shift approach as adaptive group normalization for consistency with Gupta & Brandstetter (2023).

In Table 7, we compare SINENET-8 to U-NET-MOD using both the additive (ADD) and adaptive group normalization (ADAGN) approaches on the conditional task. Models are trained on a train/valid/test split of 2,496/95/608, where all trajectories have 56 time steps and the buoyancies $f$ appearing in each split are distinct from those appearing in the remaining splits. Following Gupta & Brandstetter (2023), we evaluate each of the models on 1-step prediction with $\Delta_t \in \{0.375, 0.75, 1.5, 3.0, 6.0\}$. We additionally report errors on rollouts of length 10 with $\Delta_t = 0.375$ beginning from all possible time steps in each trajectory. Consistent with Gupta & Brandstetter (2023), we find the ADAGN versions of SineNet and U-NET-MOD to outperform the ADD versions. SINENET8-ADAGN is the top performer in all metrics.

Table 8: Results on INS for SINENET-8, SINENET-NEURAL-ODE, and SINENET-8 with reduced time history $h$.

| METHOD | # PAR. (M) | 1-STEP (%) | ROLLOUT (%) |
| --- | --- | --- | --- |
| SINENET-8 | 35.5 | 1.66 | 19.25 |
| SINENET-NEURAL-ODE | 12.7 | 1.83 | 19.88 |
| SINENET-8, $h = 1$ | 35.5 | **1.62** | **18.86** |
| SINENET-8, $h = 2$ | 35.5 | 1.64 | 19.07 |
| SINENET-8, $h = 3$ | 35.5 | 1.62 | 18.90 |

## F.2 CONNECTION TO NEURAL ODE

Neural ODE (Chen et al., 2018) is a general neural framework for learning the mapping from an initial latent representation $\boldsymbol{h}(0)$ to an output latent representation $\boldsymbol{h}(1)$ by parameterizing the derivative of $\boldsymbol{h}$ with a neural network $v_\theta$ as

$$\frac{d\boldsymbol{h}(\tau)}{d\tau} = v_\theta(\boldsymbol{h}(\tau), \tau).$$

To map from $\boldsymbol{h}(0)$ to $\boldsymbol{h}(1)$, the derivative can then be integrated as

$$\boldsymbol{h}(1) = \boldsymbol{h}(0) + \int_0^1 v_\theta(\boldsymbol{h}(\tau), \tau)d\tau,$$

where the integral is approximated using a numerical integrator. Chen et al. (2018) derive a method for backpropagating through the operations of an arbitrary numerical integrator in a stable and memory-efficient manner, enabling training of $v_\theta$. Their method includes the case of adaptive time-stepping integrators, which evaluate the derivative $\frac{d\boldsymbol{h}(\tau)}{d\tau}$ an adaptive number of times dependent on $\boldsymbol{h}(\tau)$ to induce stability in the integration. This allows $v_\theta$ to be applied a variable number of times in mapping $\boldsymbol{h}(0)$ to $\boldsymbol{h}(1)$, thereby formulating a continuous-depth neural network.

A number of common machine learning tasks fit in this framework, including image classification, image generation, and time series modeling. Taking $\boldsymbol{h}(\tau)$ as the latent solution $\boldsymbol{x}_{t+\tau}$ and $v_\theta$ as a U-Net frames the mapping from $\boldsymbol{x}_t$ to $\boldsymbol{x}_{t+1}$ as a neural ODE formulation of SineNet which we refer to as SINENET-NEURAL-ODE. We train and evaluate SINENET-NEURAL-ODE on INS using the fifth order Dormand-Prince-Shampine numerical integrator, which is the default in the Neural-ODE PyTorch library (Chen, 2018). Following the MNIST (LeCun et al., 1998) experiment from Chen et al. (2018), we reduce the parameters of SINENET-NEURAL-ODE roughly by a factor of 3 relative to SineNet. We incorporate the coordinate $\tau$ via a learned embedding vector added along the channel dimension in each convolution block of $v_\theta$ as done in the conditional task in Appendix F.1.

Although Chen et al. (2018) showed that $v_\theta$ could be optimized with space complexity independent of the number of evaluations, we instead train SINENET-NEURAL-ODE with direct back propagation to speed up training. As shown in Figure 7, the number of function evaluations for SINENET-NEURAL-ODE increases as training progresses, which is consistent with the findings of Chen et al. (2018). Due to the high number of U-Net calls per forward pass, SINENET-NEURAL-ODE is expensive to train in terms of space and time despite the reduction in the number of parameters, as we show in Table 4. We present test results for SINENET-NEURAL-ODE in Table 8, where we find its performance to be competitive with SINENET-8.

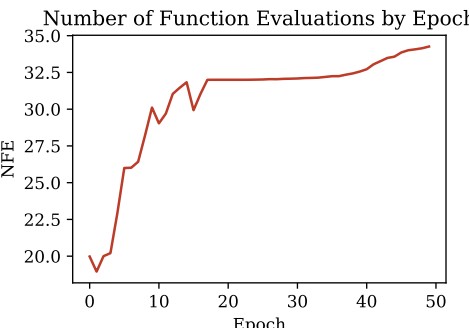

Figure 7: Average number of forward passes per batch by epoch when training SINENET-NEURAL-ODE. Consistent with Chen et al. (2018), we find that the number of forward passes increases as training progresses.

### F.3 SUPER-RESOLUTION ANALYSIS

Table 9: Summary of rollout test error and one-step test error for the super-resolution task on 1,000 CNS trajectories downloaded from PDEBench (Takamoto et al., 2022).

| METHOD | # PAR. (M) | $128 \times 128$ 1-STEP (%) | ROLLOUT (%) | SR METHOD | $512 \times 512$ 1-STEP (%) | ROLLOUT (%) |
|---|---|---|---|---|---|---|
| SINENET-8 | 35.5 | **1.06** | **2.64** | DILATION | **1.06** | **2.65** |
| | | | | INTERPOLATION | 2.44 | 3.54 |
| F-FNO | 30.1 | 1.46 | 2.93 | DIRECT | 1.44 | 2.90 |

As discussed in Section 2, neural operators (Kovachki et al., 2021) aim to learn PDE solution operators independently of the resolution of the training data. This enables generalization beyond the discretization of the training data such that a trained neural operator can perform *zero-shot super resolution* (Li et al., 2021a), wherein the task is to solve the PDE at a higher resolution than during training. As U-Nets, SineNets, and other CNN-based architectures learn their kernel functions on the same grid as the training data, they cannot perform this task directly, although recent work has adapted CNNs to the neural operator framework (Raonić et al., 2023).

To perform super-resolution with SineNet, we consider two approaches: **interpolation** and **dilation**.

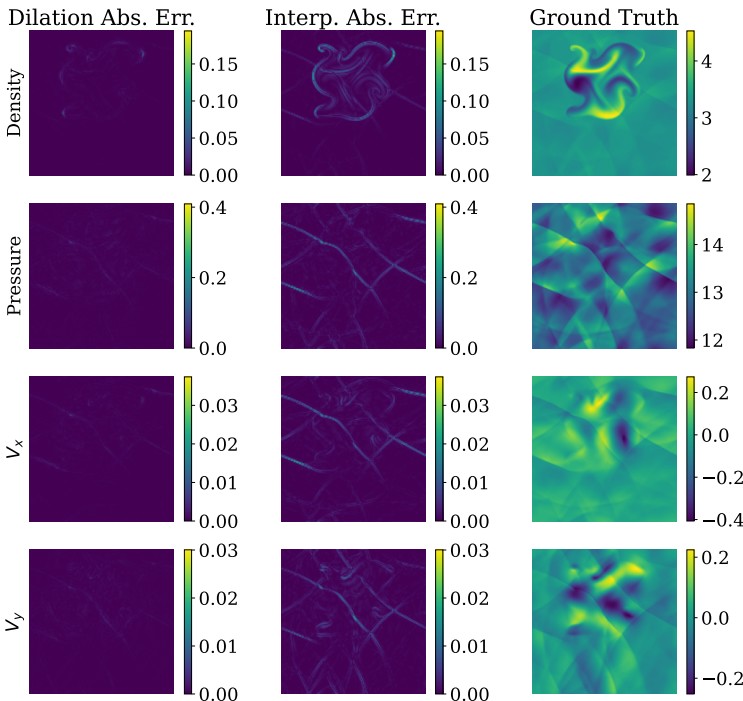

Figure 8: Absolute error for super-resolved one-step prediction on CNS using dilation and interpolation.

**Interpolation.** The input initial 10 time steps are downsampled to the training resolution of $128 \times 128$. SineNet then solves the PDE at the lower resolution, after which the solution is interpolated to the higher resolution.

**Dilation.** Alternatively, SineNet can operate directly on the high resolution data using dilation. Convolutions, downsampling, and upsampling operations from the SineNet trained on $128 \times 128$ are all dilated by a factor of 4. Intuitively, the grid can be viewed to be divided using a checkerboard pattern, where each grid point interacts exclusively with other grid points that share the same type (color) as designated by this checkerboard arrangement. This ensures that even at a higher resolution, each feature map grid point interacts only with grid points spaced equidistant to those it would interact with at the training resolution. Although dilation is a standard operation in convolution layers, we highlight that it is crucial to also apply dilation in pooling (for downsampling) and in interpolation (for upsampling). For example, in the $2 \times 2$ pooling operation, instead of averaging and reducing over $2 \times 2$ regions of immediate neighboring grid points, we average and reduce over the 4 corner grid points of $5 \times 5$ regions.

We download 1,000 $512 \times 512$ CNS trajectories from PDEBench (Takamoto et al., 2022) and evaluate models trained on the $128 \times 128$ CNS data presented in the main text. We compare to F-FNO, a neural operator which does not require interpolation or dilation to perform super-resolution. Results are presented in Table 9. As in the main text, SINENET-8 outperforms F-FNO at training resolution. At the higher resolution, interpolation introduces error, while dilation achieves nearly identical error to at training resolution. In Figure 8, we visualize the super-resolved errors using both approaches for one-step prediction on a randomly chosen example. As can be seen, unlike dilation, interpolation introduces errors in regions of high gradient. However, dilation is only applicable for super-resolving at integer multiples of the training resolution. For non-integer super-resolution multiples, similar techniques in deformable convolutions (Dai et al., 2017) could be considered to offset the operations to non-integer locations during testing. Furthermore, all super-resolution approaches introduce error in settings where the higher resolution solutions contain high-frequency details not present at training resolution, or when numerical simulation at a higher resolution alters the behavior of the dynamics relative to training resolution, *e.g.*, by introducing smaller eddies in a fluid simulation.

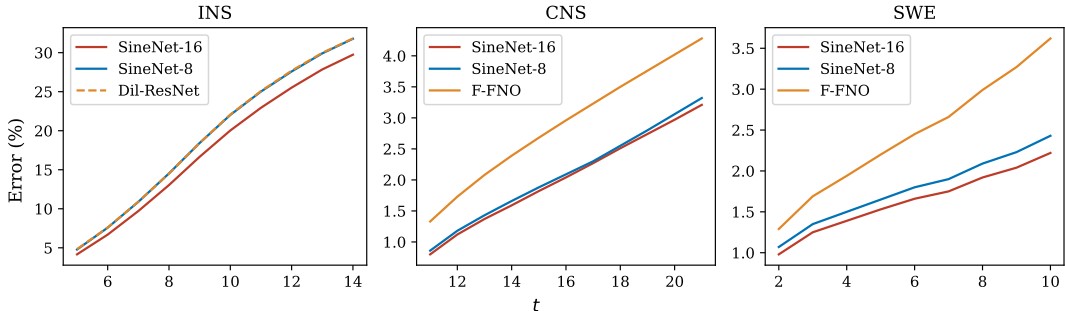

Figure 9: Evolution of rollout error for SINENET-8, SINENET-16, and the best baseline on each dataset.

### F.4 TEMPORAL EVOLUTION OF ROLLOUT ERROR

In Figure 9, we visualize the evolution of the rollout error of SineNet against the best baseline on each of the considered datasets.

### F.5 LONG-TIME PREDICTION ON CNS

To evaluate SineNet on longer time horizons, we generated 100 CNS trajectories with $T = 120$ time steps. As opposed to the CNS task considered in the main text where solvers unroll trajectories for 11 steps given the initial 10 steps, we increase the time horizon by a factor of 10 in unrolling 110 steps given the initial 10 steps. Although the initially turbulent dynamics stabilize over the lengthy trajectory and therefore present a more stable prediction target, we evaluated models trained on the original CNS task such that beyond $t = 21$, the dynamics were out-of-distribution.

One approach to increase long-term stability is noise injection (Sanchez-Gonzalez et al., 2020; Stachenfeld et al., 2022), where 0-mean Gaussian noise is added to training inputs to simulate noise encountered by the solver during rollouts due to errors in previous predictions. Recent work by Lippe et al. (2023) extended this approach to consider multiple levels of noise that the model is trained to remove, which shares a connection with denoising diffusion probabilistic models (Ho et al., 2020). Tran et al. (2023) found noise injection to be important for stable training of F-FNO, which we employed in training F-FNO for our tasks. For this longer rollout task, we additionally evaluated a SINENET-8-NOISE trained on the 21-step CNS data with the noise level $\sigma = 0.01$, which is the same as F-FNO.

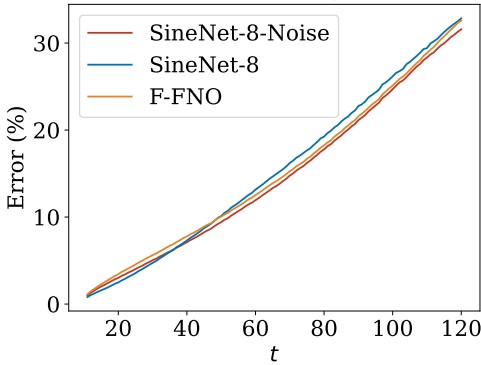

Figure 10: Rollout error averaged across 89 CNS trajectories of length $T = 120$ for SINENET-8-NOISE, SINENET-8, and F-FNO.

We present results in Table 10. We find a subset of the generated trajectories led to poor performance across all models, even causing several of the baselines to diverge entirely and produce solutions with greater than 100% error in 11/100 cases. The cause is unclear, although the initial density and/or pressure fields for each of these trajectories have small mean values relative to the remaining trajectories, potentially creating out-of-distribution dynamics. We therefore present results both with and without these 11 trajectories. SINENET-8-NOISE has the lowest rollout error in both cases, and is more robust to the 11 difficult cases than the remaining models.

In Figure 10, we present the rollout error by time step for SINENET-8 trained with and without noise, as well as F-FNO. As can be seen, noise injection has a regularizing effect that causes the predictions of SINENET-8 to have better errors for short time horizons than models trained with noise injection. However, at longer time horizons, models trained with noise injection surpass SINENET-8 due to

their robustness to the accumulation of rollout error. In Figures 11-14, we visualize a randomly selected trajectory predicted by SINENET-8-NOISE on this data.

Table 10: Summary of rollout test errors on CNS with $T = 120$. ROLLOUT-89 corresponds to the dataset with the 11 trajectories resulting in a baseline error greater than 100% removed, while ROLLOUT-100 is for the full dataset. '–' indicates a rollout error greater than 100%.

| METHOD | # PAR. (M) | ROLLOUT-100 (%) | ROLLOUT-89 (%) |
|---|---|---|---|
| SINENET-8-NOISE | 35.5 | **15.94** | **14.62** |
| SINENET-8 | 35.5 | 27.97 | 15.44 |
| F-FNO | 30.1 | – | 15.15 |
| DIL-RESNET | 16.7 | – | 23.44 |
| U-NET-128 | 135.1 | 29.97 | 15.25 |
| U-NET-MOD | 144.3 | – | 18.95 |

## F.6 ABLATION ON NUMBER OF CONDITIONING STEPS

In Table 8, we ablate the number of historical conditioning steps used on INS. We find that $h = 1$ gives the lowest test error, followed by $h = 3$. That $h = 1$ performs as well or better than $h > 1$ is in line with findings reported by Tran et al. (2023). In our experiments, we instead adhere to the number of historical steps used by the respective benchmarks on each dataset (4 for INS, 10 for CNS and 2 for SWE). However, we believe that principled approaches for choosing the number of conditioning steps, as well as how to incorporate them into model predictions, is an interesting topic for future research.

## F.7 WAVE BOTTLENECK

Table 11: Results on INS, CNS, and SWE for SINENET-8 and SINENET-8-BOTTLENECK.

| METHOD | # PAR. (M) | INS 1-STEP (%) | INS ROLLOUT (%) | CNS 1-STEP (%) | CNS ROLLOUT (%) | SWE 1-STEP (%) | SWE ROLLOUT (%) |
|---|---|---|---|---|---|---|---|
| SINENET-8 | 35.5 | **1.66** | 19.25 | **0.93** | **2.10** | 1.02 | **1.78** |
| SINENET-8-BOTTLENECK | 35.5 | 1.68 | **19.19** | 1.20 | 3.14 | 1.63 | 2.51 |

In addition to the dual processing mechanism which we discuss in Section 3.5 and ablate with SINENET-8-ENTANGLED in Section 5.4, a primary difference between the waves $V_k$ comprising SineNet and a conventional U-Net is the encoder and decoder present in conventional U-Nets ($P$ and $Q$ in Equations 1 and 2). SineNet instead maintains a high-dimensional representation between waves. We ablate this design choice with SINENET-8-BOTTLENECK, which decodes the latent solution $x_{t+\Delta_k}$ output by $V_k$ to the lower-dimensional base space before encoding back to the latent space for input to $V_{k+1}$. Aside from the dual processing mechanism, this renders the architecture of each wave $V_k$ closer to that of a conventional U-Net, however, it creates a bottleneck between waves.

In Table 11, we present results for SINENET-8-BOTTLENECK evaluated on INS, CNS, and SWE. On all datasets, SINENET-8 outperforms SINENET-8-BOTTLENECK in terms of 1-step error. While SINENET-8 has a lower rollout error than SINENET-8-BOTTLENECK on CNS and SWE by a substantial margin, SINENET-8-BOTTLENECK tops the rollout error of SINENET-8 on INS.

The mixed results on these datasets could be due to the difficulty of the initial time steps in the INS rollouts relative to the remaining time steps. Compared to the 1-step error of 1.66 for SINENET-8 averaged over all possible time steps in INS trajectories, Figure 9 shows that the errors starting from the beginning of the trajectory are over twice as high. This is likely due to the large velocities in earlier timesteps creating dynamics which are faster-evolving and with a greater degree of local variability than at later timesteps, which is exemplified in Figures 18-20. As a result of larger errors early on in predicted trajectories, as well as the accumulation of error through autoregressive rollout, there is a substantially larger gap between 1-step errors and rollout errors on INS. We hypothesize that the bottleneck serves as a form of regularization which decreases performance in terms of 1-step prediction, but increases robustness to the difficult initial steps in INS. Performing a similar analysis

for CNS and SWE, the difficulty of the initial timesteps instead appear similar to the remaining timesteps on both datasets. Thus, the regularization is not beneficial as before, and in fact leads to a performance drop, likely due to the information bottleneck between waves.

We visualize the decoded feature maps between each wave for SINENET-8-BOTTLENECK on INS, CNS, and SWE in Figures 15, 16, and 17, respectively. Interestingly, as SWE is a global weather forecasting task, we can see the outline of the world map in the visualized feature maps (*e.g.,* Feature Map 1 of Wave 3 in Figure 17), which implies that the primary objective of these feature maps is for modeling the evolution of dynamics about the boundaries of continents, potentially encouraged by the information compression in the wave bottleneck.

### F.8    SINENET-$K$ RESULTS

In Table 12, we present numerical results for the SineNets with varying $K$ visualized in Figure 3.

Table 12: Summary of rollout test error and one-step test error for SineNet with varying $K$.

| METHOD | # PAR. (M) | INS 1-STEP (%) | INS ROLLOUT (%) | CNS 1-STEP (%) | CNS ROLLOUT (%) | SWE 1-STEP (%) | SWE ROLLOUT (%) |
|---|---|---|---|---|---|---|---|
| SINENET-2 | 35.5 | 2.38 | 23.71 | 1.56 | 4.98 | 1.38 | 2.54 |
| SINENET-4 | 35.5 | 1.91 | 21.06 | 1.26 | 2.63 | 1.13 | 1.97 |
| SINENET-6 | 35.5 | 1.73 | 19.58 | 1.12 | 2.40 | 1.10 | 1.91 |
| SINENET-8 | 35.5 | 1.66 | 19.25 | 0.93 | 2.10 | 1.02 | 1.78 |
| SINENET-10 | 35.5 | 1.58 | 18.68 | 0.94 | 2.24 | 0.97 | 1.73 |
| SINENET-12 | 35.5 | 1.53 | 18.09 | 0.93 | 2.12 | 0.92 | 1.64 |
| SINENET-14 | 35.4 | 1.49 | 17.66 | 0.89 | **1.97** | 0.94 | 1.66 |
| SINENET-16 | 35.5 | **1.46** | **17.63** | **0.87** | 2.04 | **0.92** | **1.64** |

## G    DATASET AND PREDICTION VISUALIZATION

Here, we visualize rollout predictions from SINENET-8 on INS (Figures 18-20), CNS (Figures 21-24), and SWE (Figures 25-27) on a randomly selected trajectory from each test set. In each, we show the ground truth field in the left column, the predicted field in the middle column, and the absolute error in the right column.

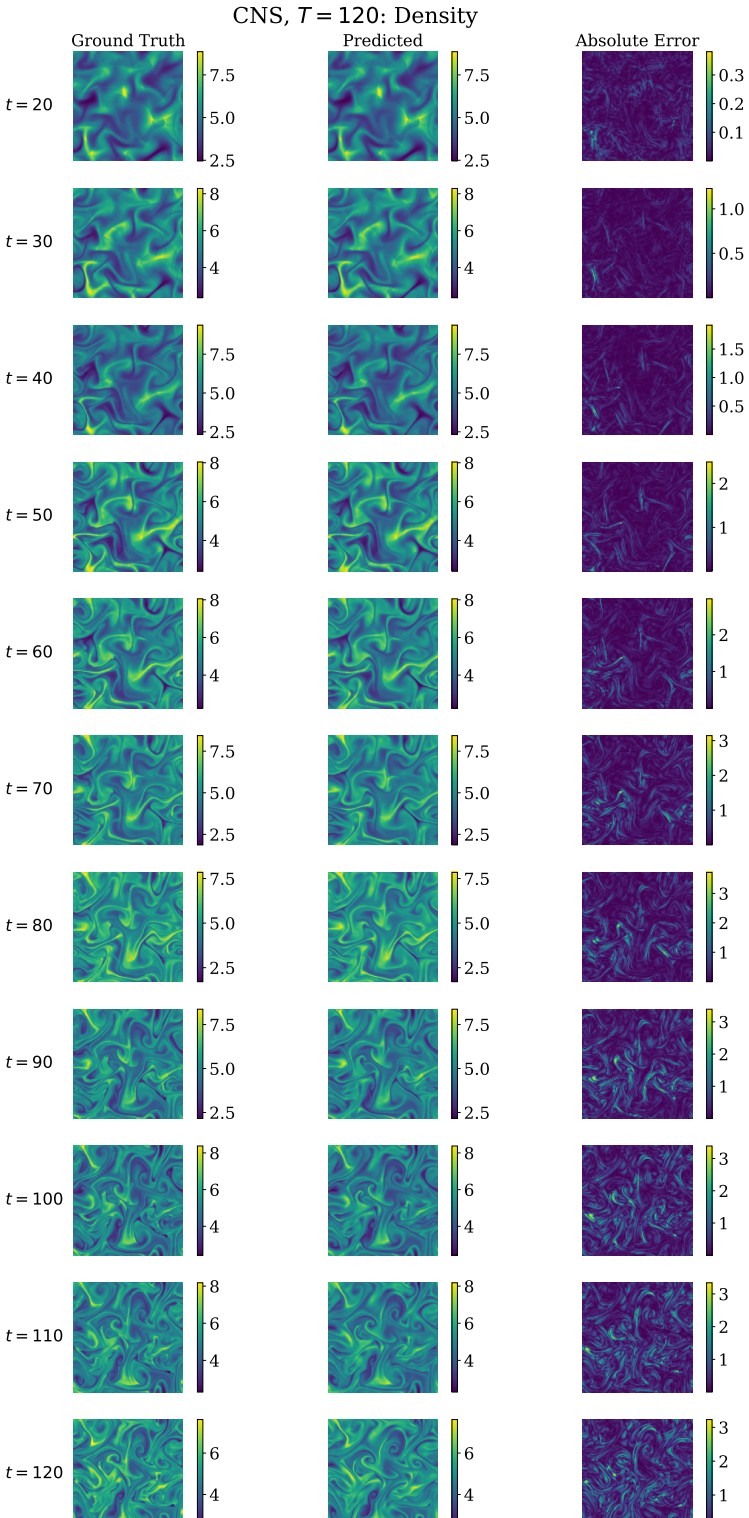

Figure 11: Compressible Navier-Stokes density field for $T = 120$ downsampled to every 10 time steps.

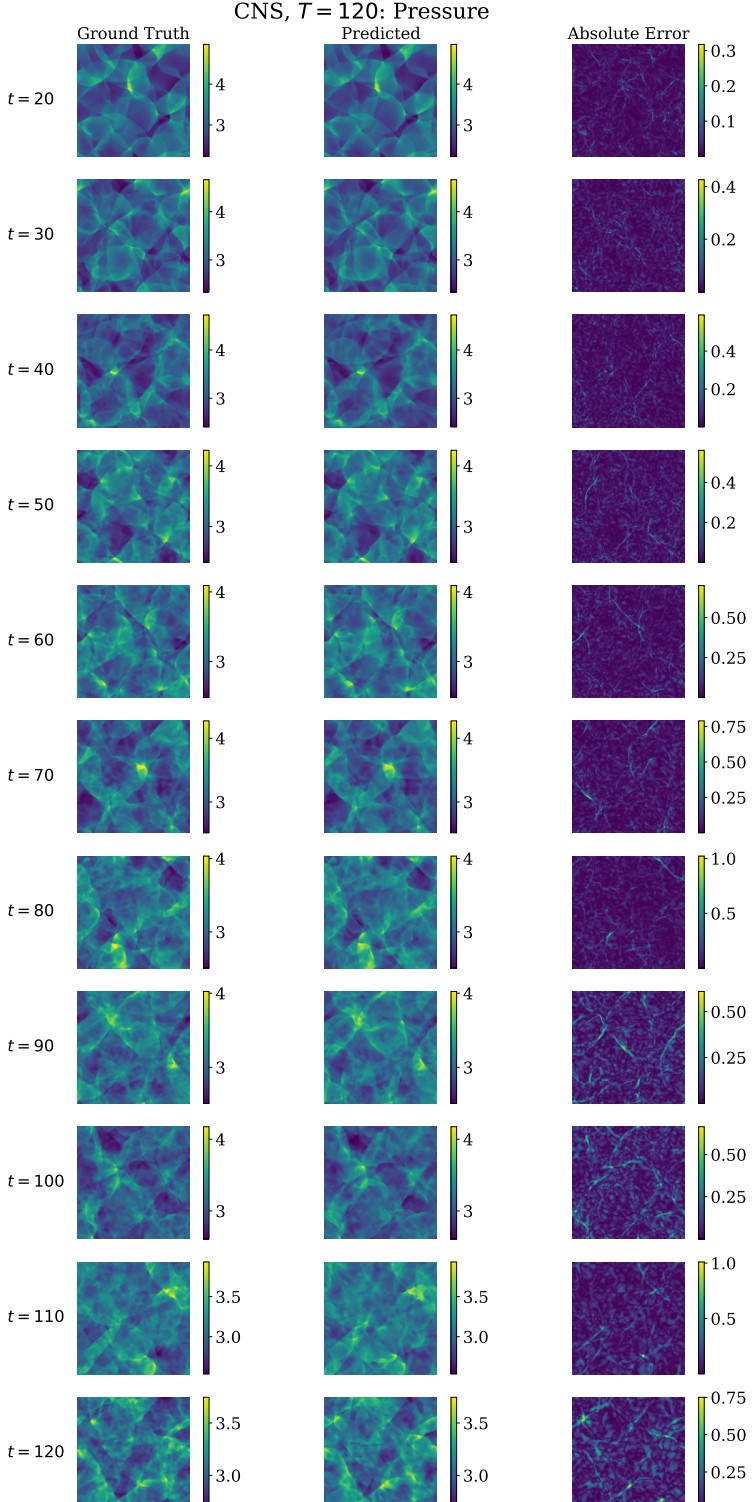

Figure 12: Compressible Navier-Stokes pressure field for $T = 120$ downsampled to every 10 time steps.

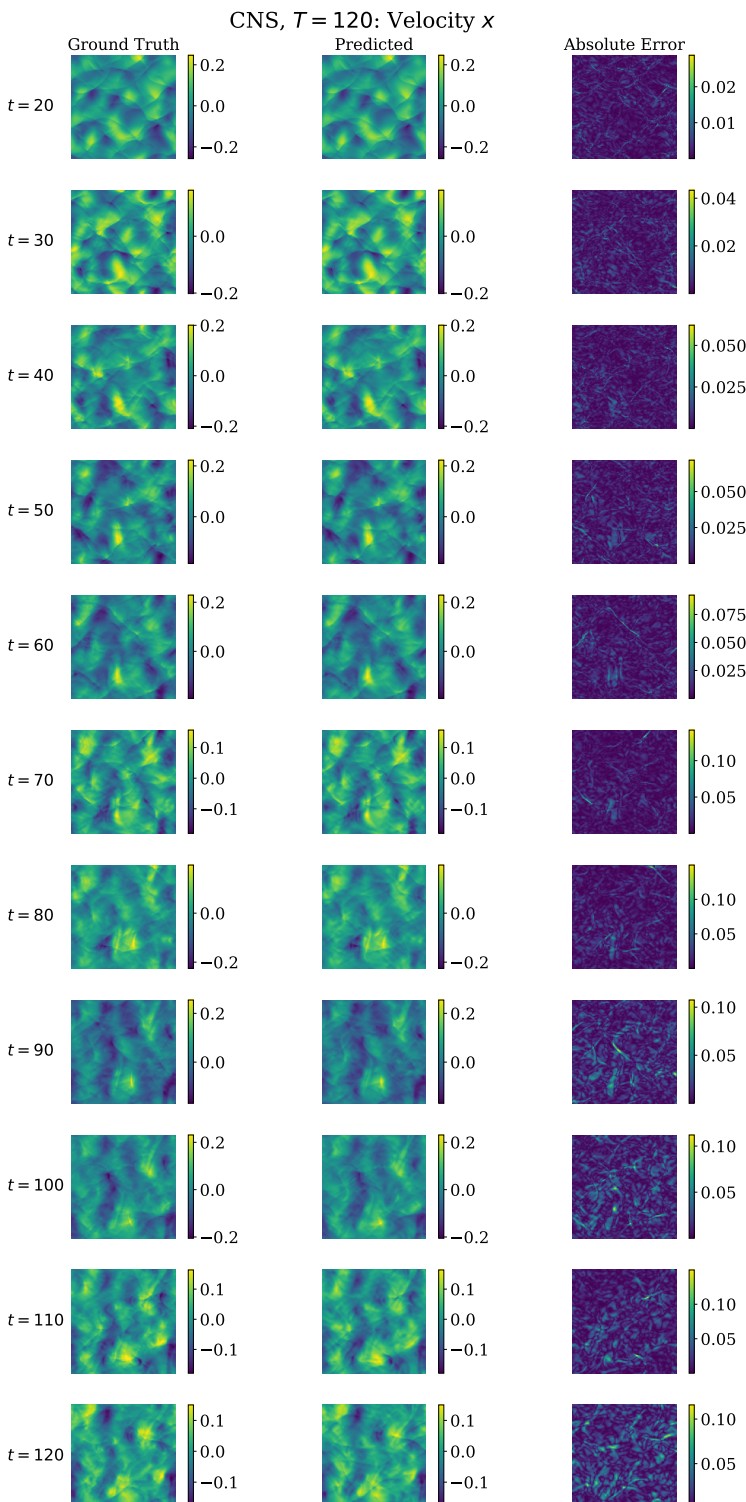

Figure 13: Compressible Navier-Stokes velocity $x$ component for $T = 120$ downsampled to every 10 time steps.

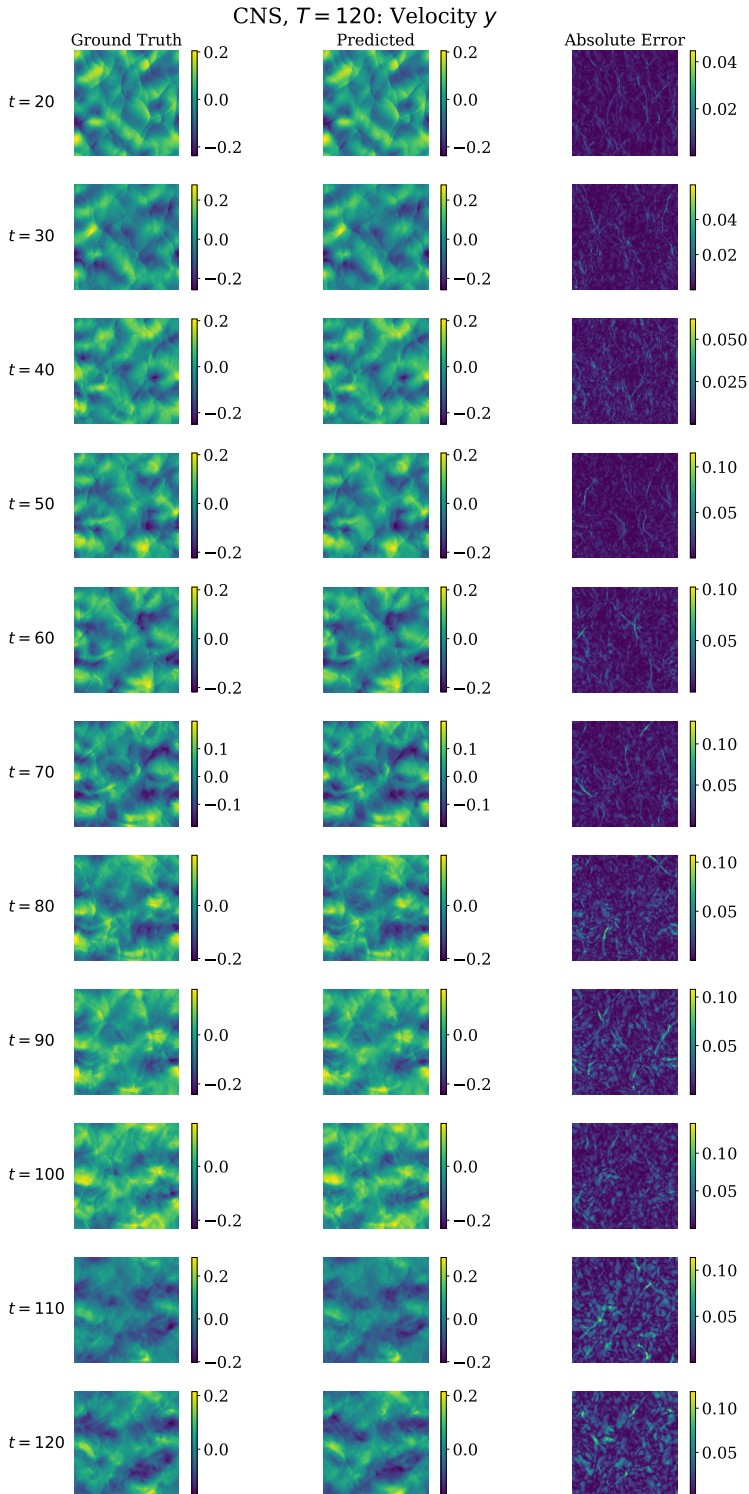

Figure 14: Compressible Navier-Stokes velocity $y$ component for $T = 120$ downsampled to every 10 time steps.

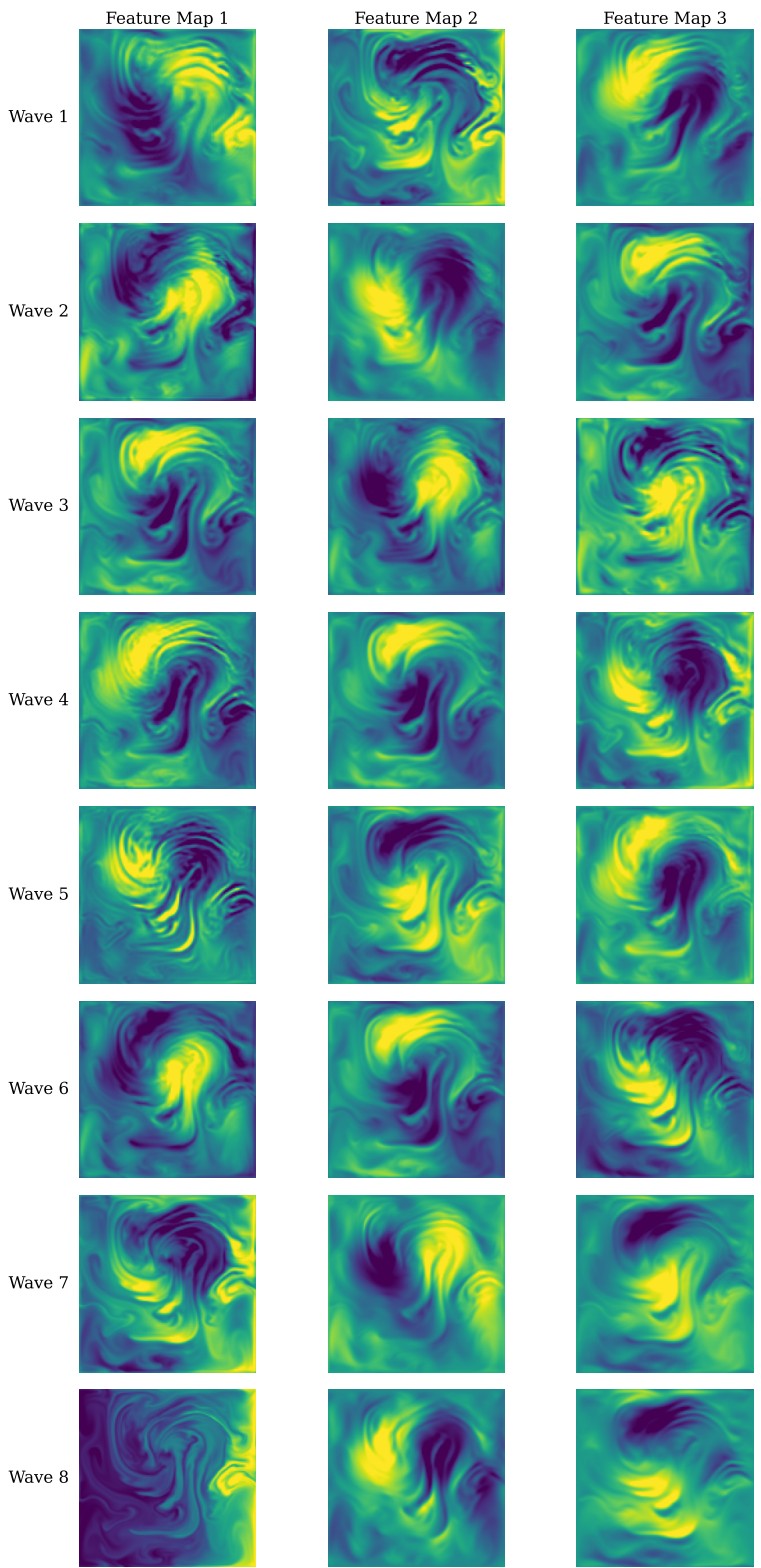

Figure 15: Visualization of SINENET-8-BOTTLENECK decoded feature maps between each wave on INS. The Wave 8 feature maps are the predicted particle concentration and velocity field.

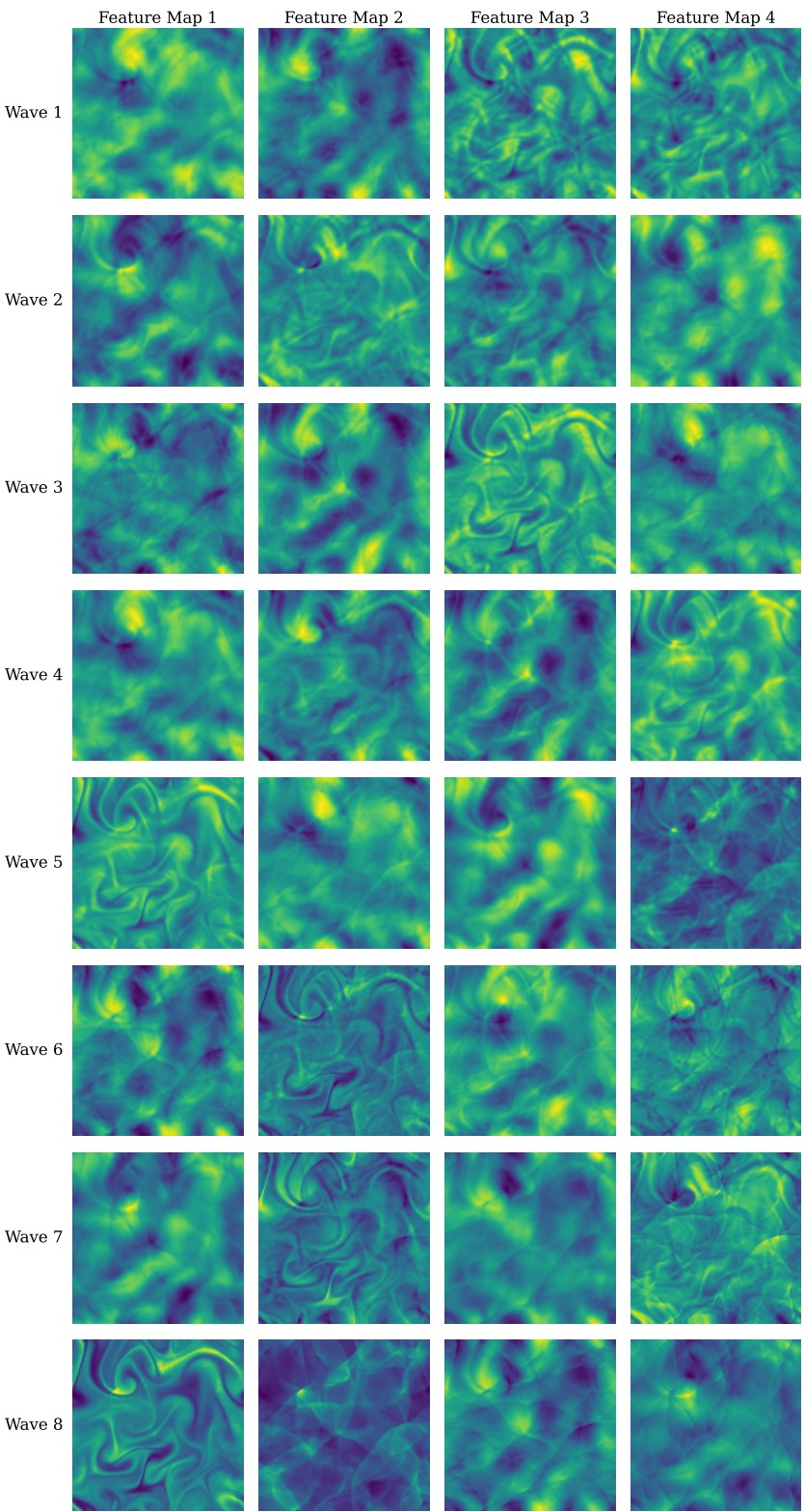

Figure 16: Visualization of SINENET-8-BOTTLENECK decoded feature maps between each wave on CNS. The Wave 8 feature maps are the predicted density, pressure and velocity fields.

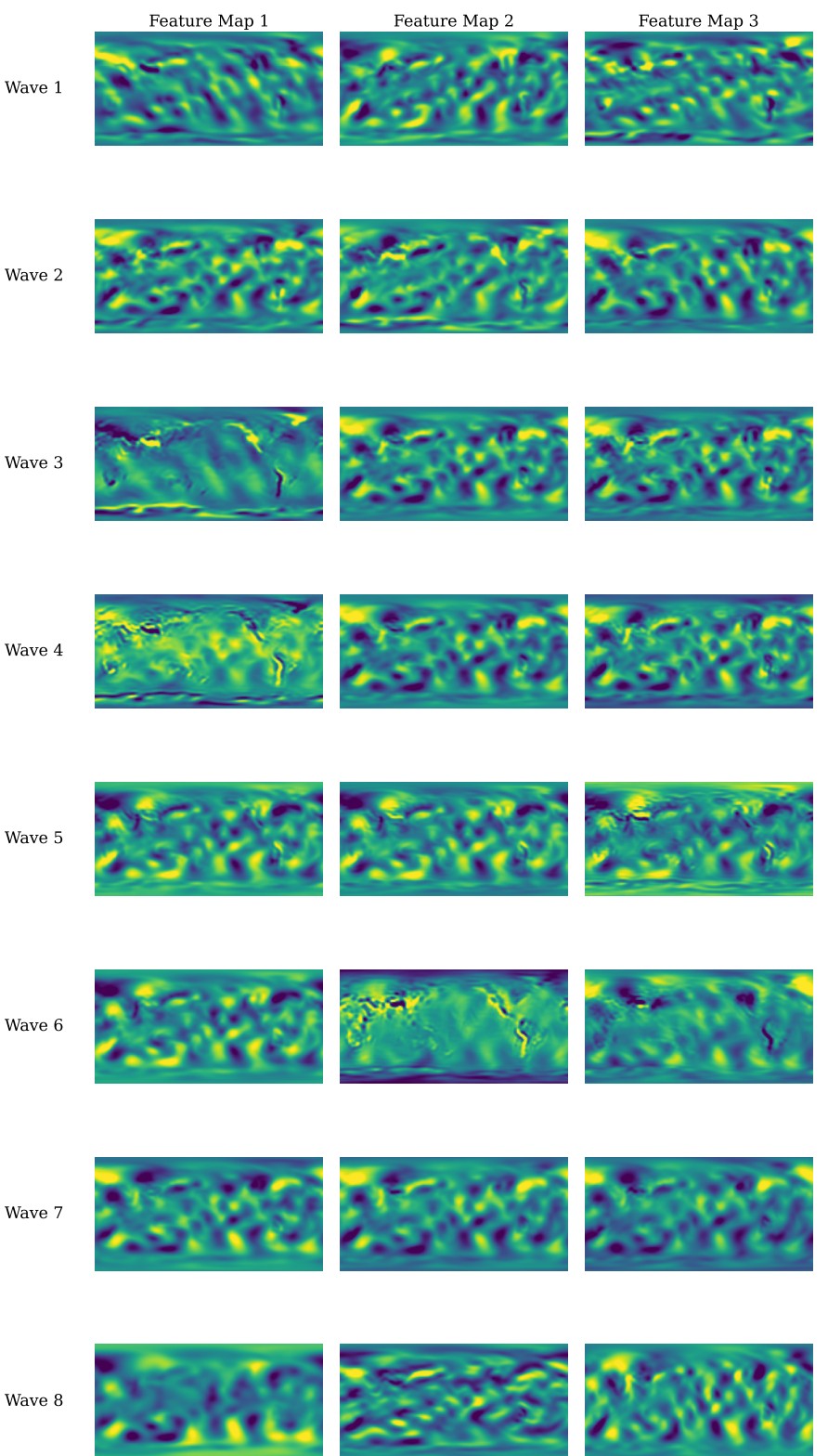

Figure 17: Visualization of SINENET-8-BOTTLENECK decoded feature maps between each wave on SWE. The Wave 8 feature maps are the predicted pressure and velocity field.

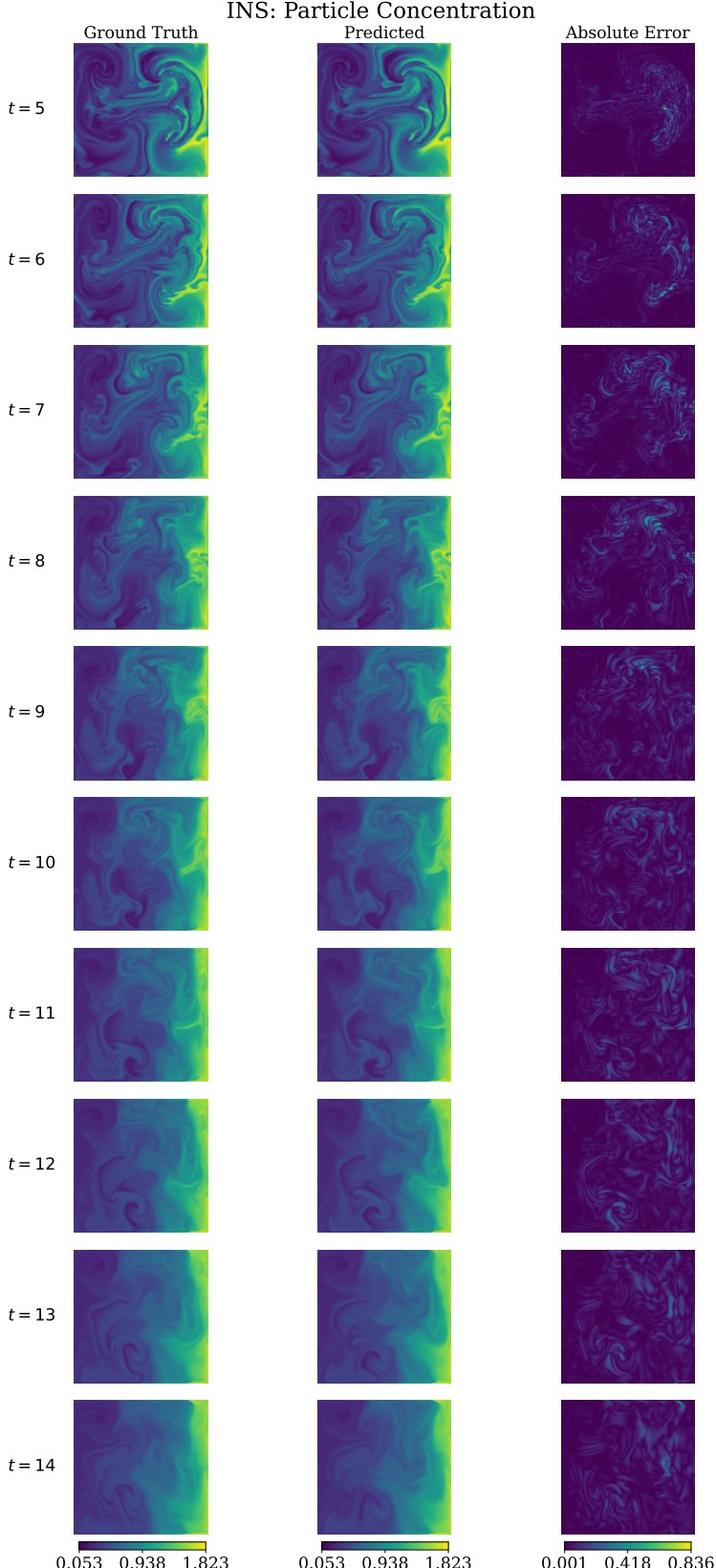

Figure 18: Incompressible Navier-Stokes particle concentration.

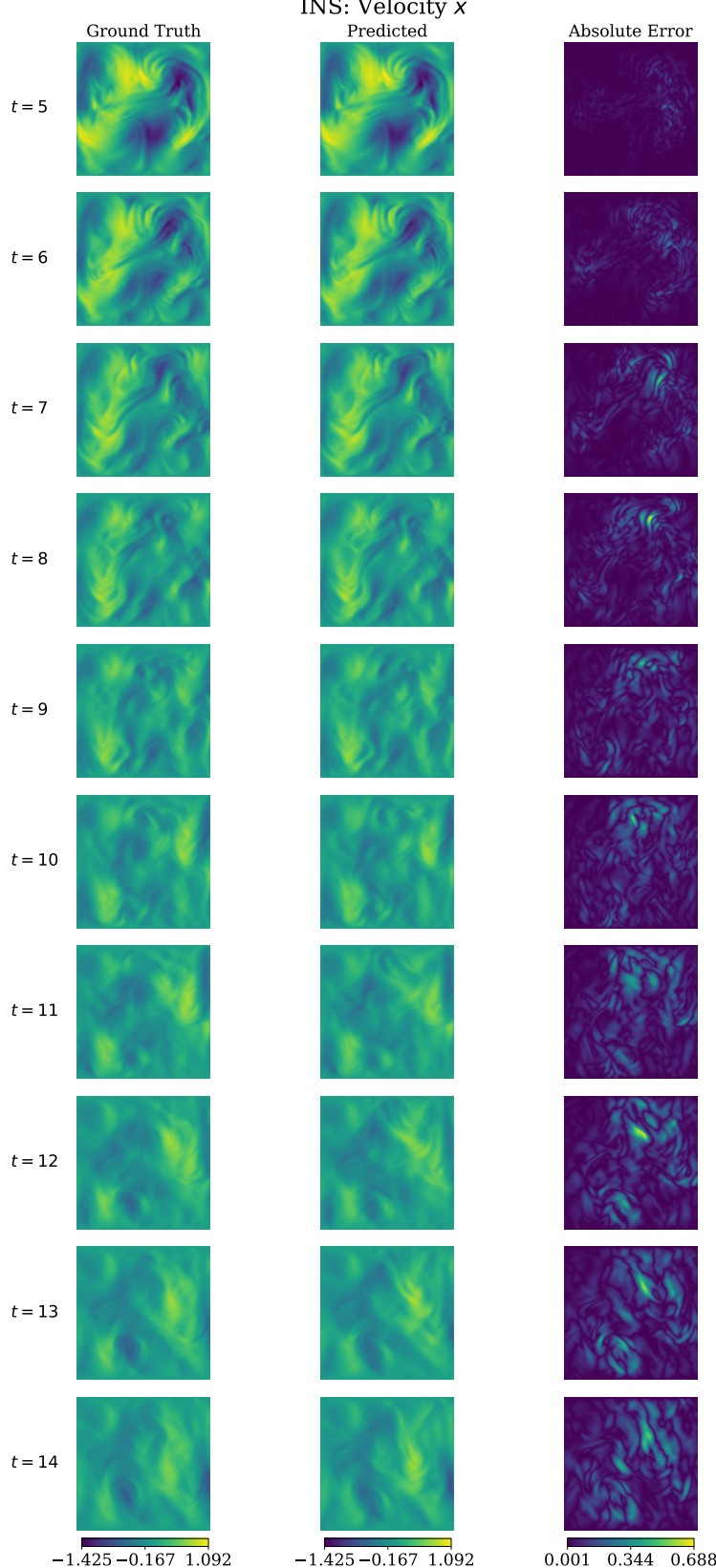

Figure 19: Incompressible Navier-Stokes velocity $x$ component.

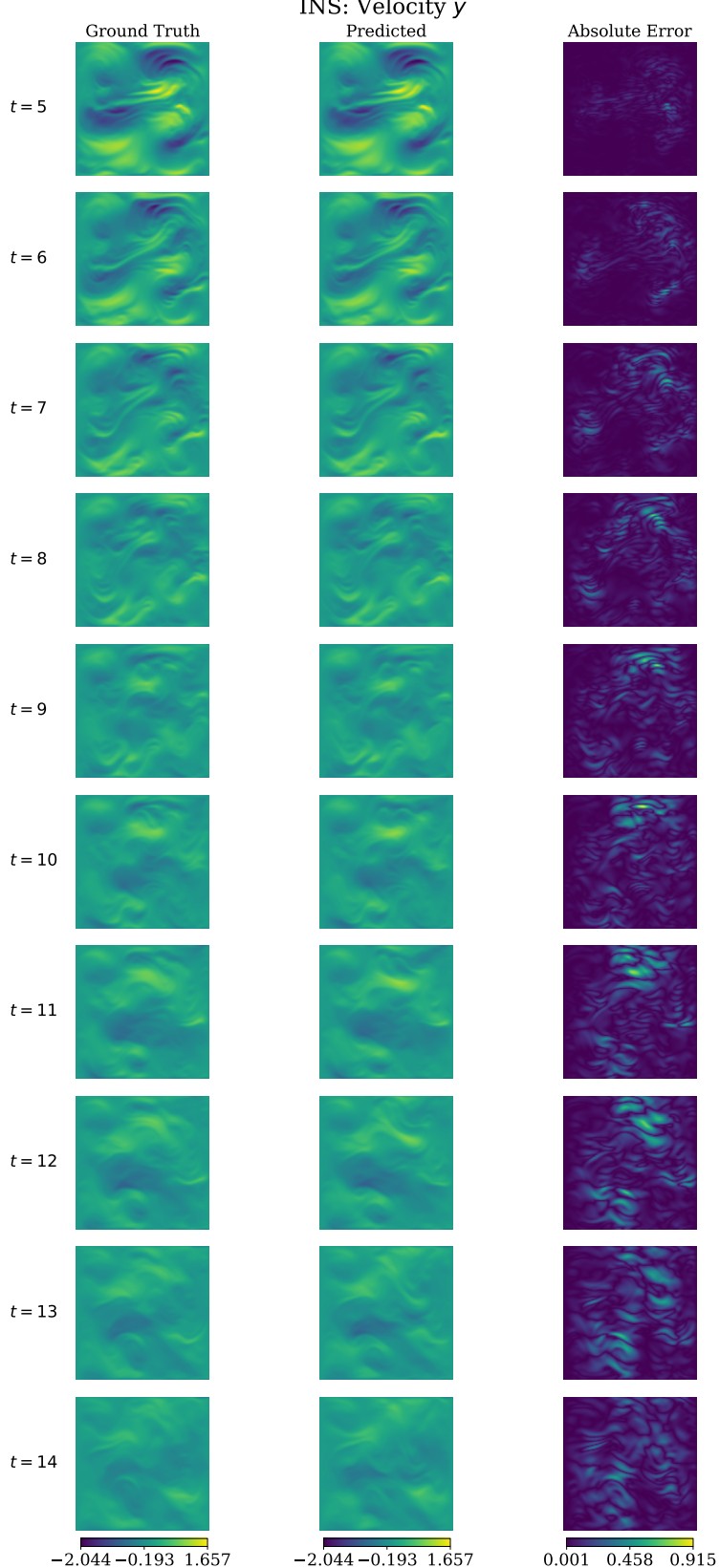

Figure 20: Incompressible Navier-Stokes velocity $y$ component.

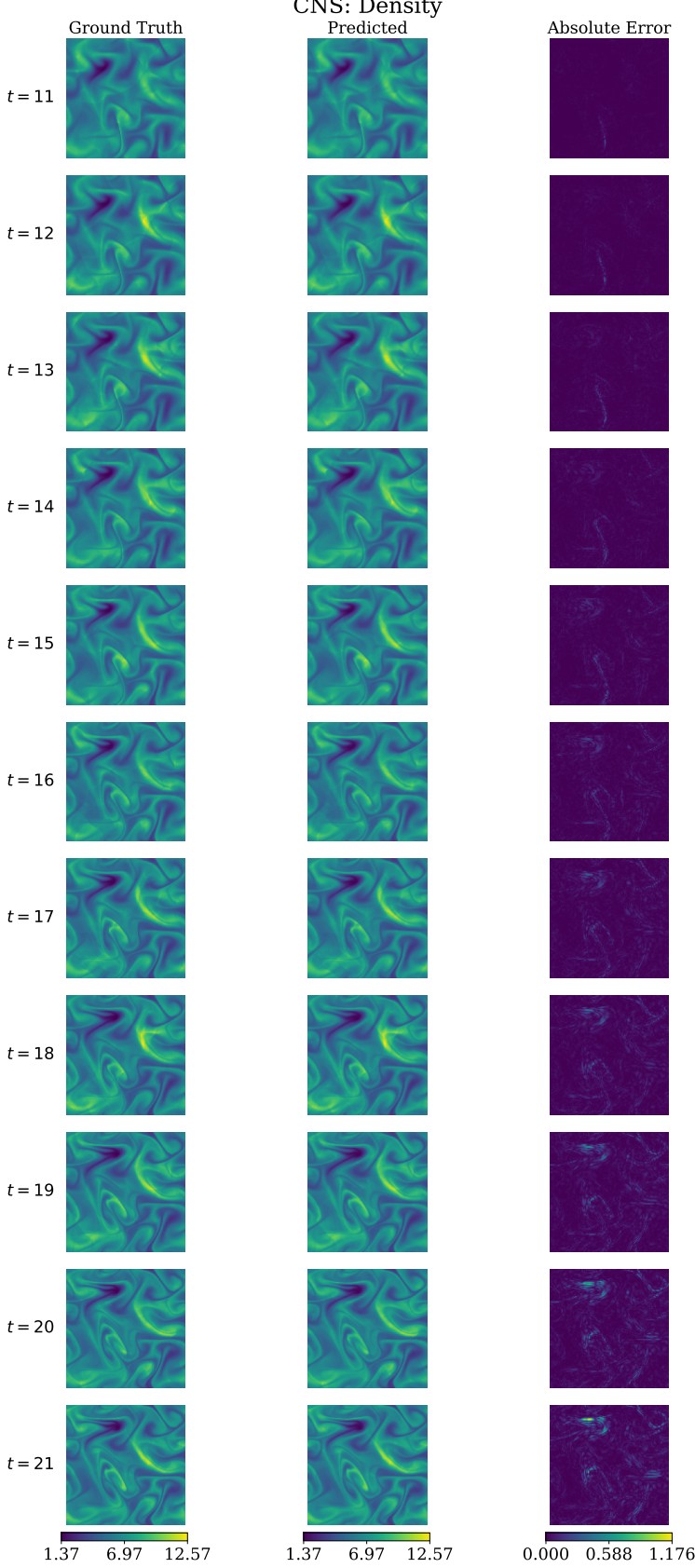

Figure 21: Compressible Navier-Stokes density field.

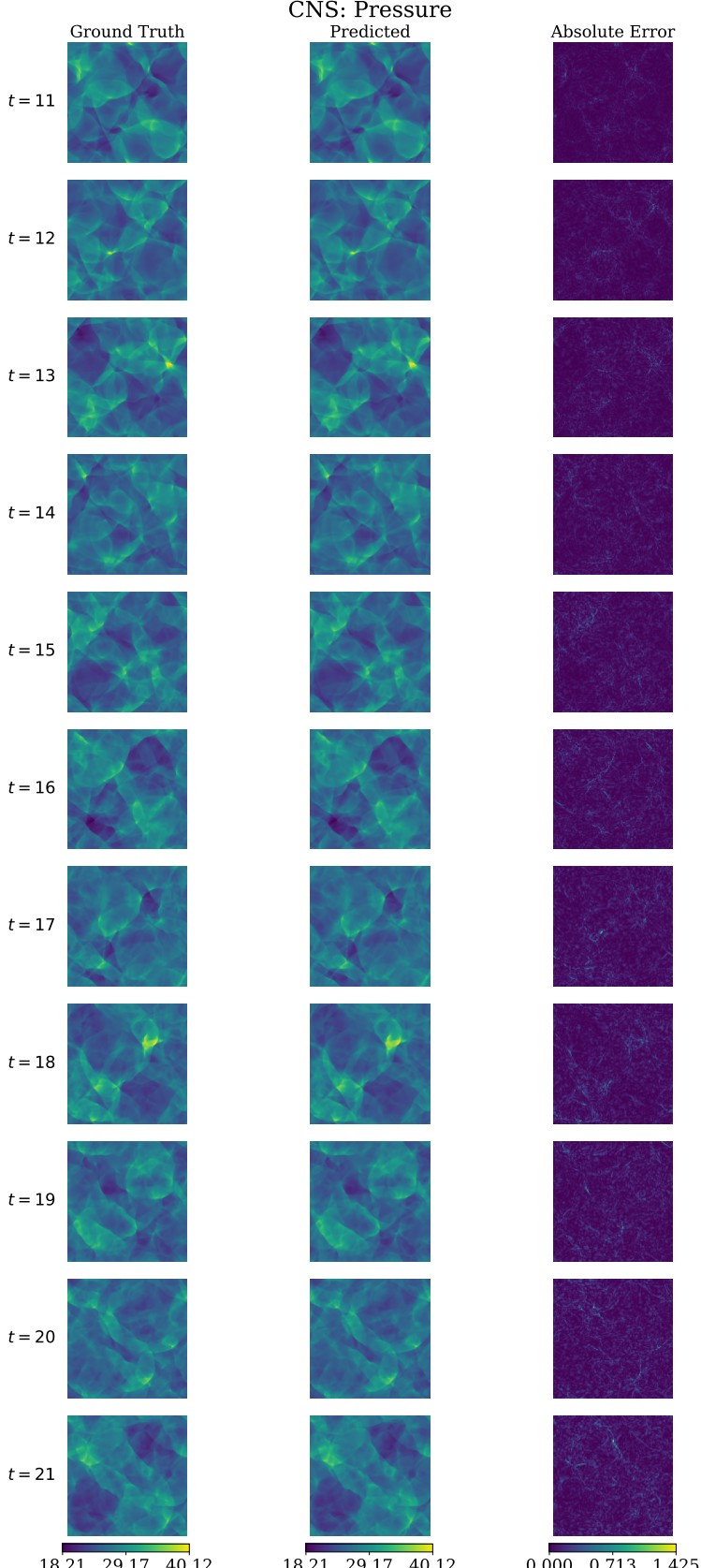

Figure 22: Compressible Navier-Stokes pressure field.

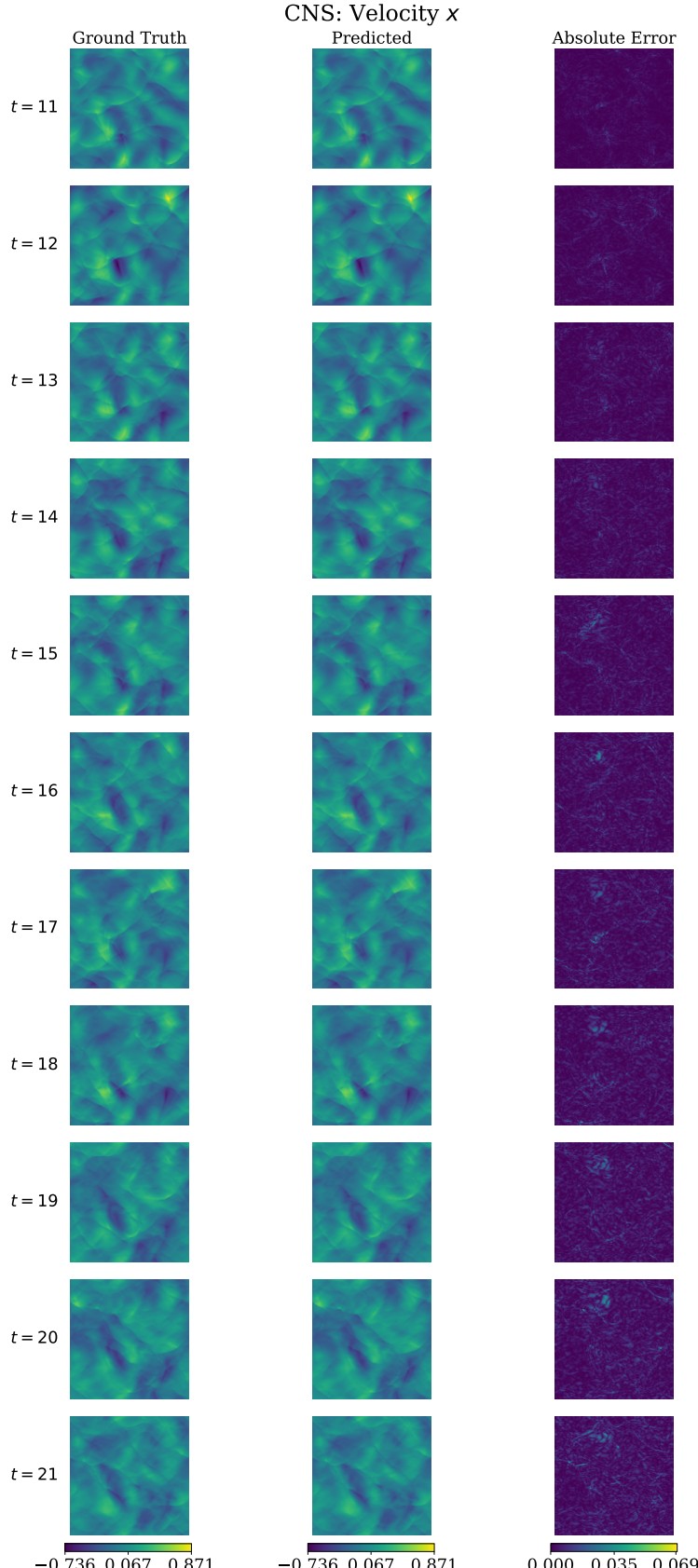

Figure 23: Compressible Navier-Stokes velocity $x$ component.

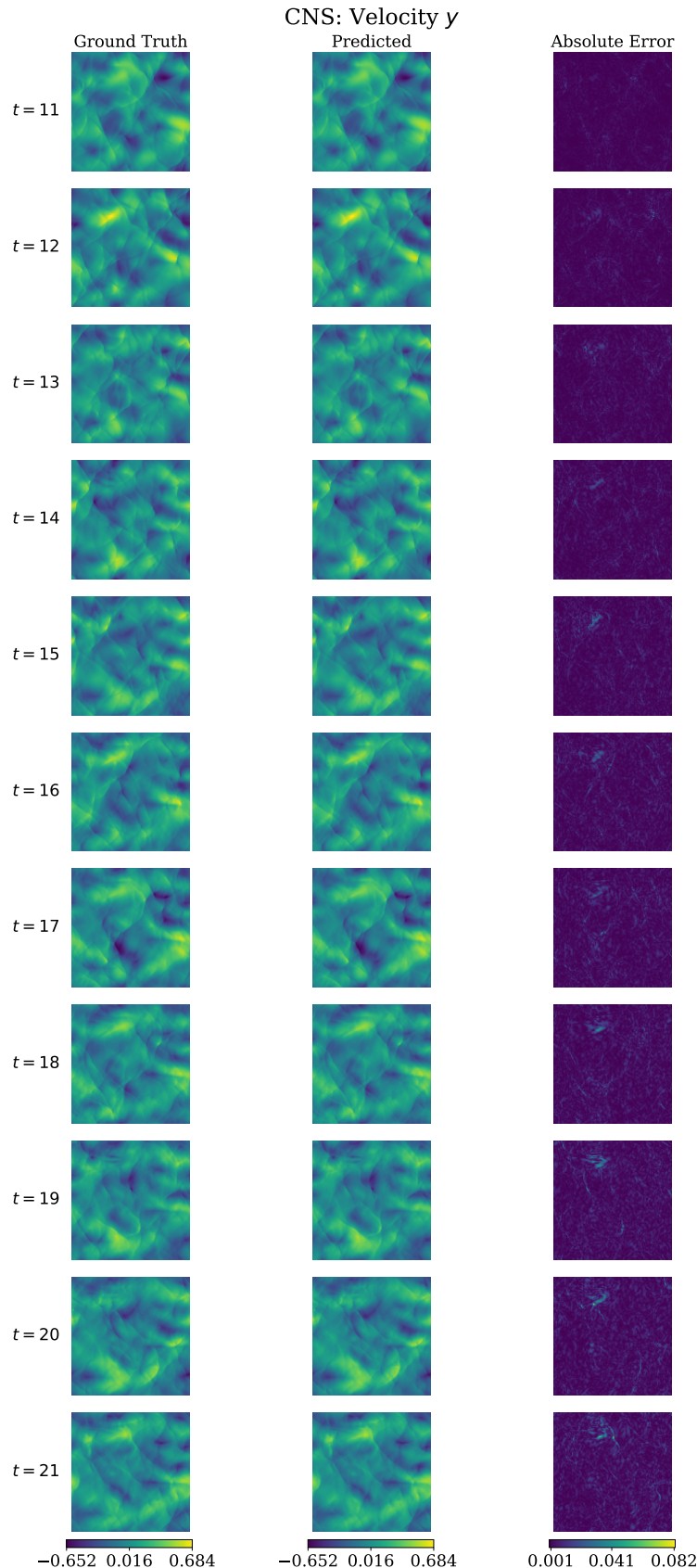

Figure 24: Compressible Navier-Stokes velocity $y$ component.

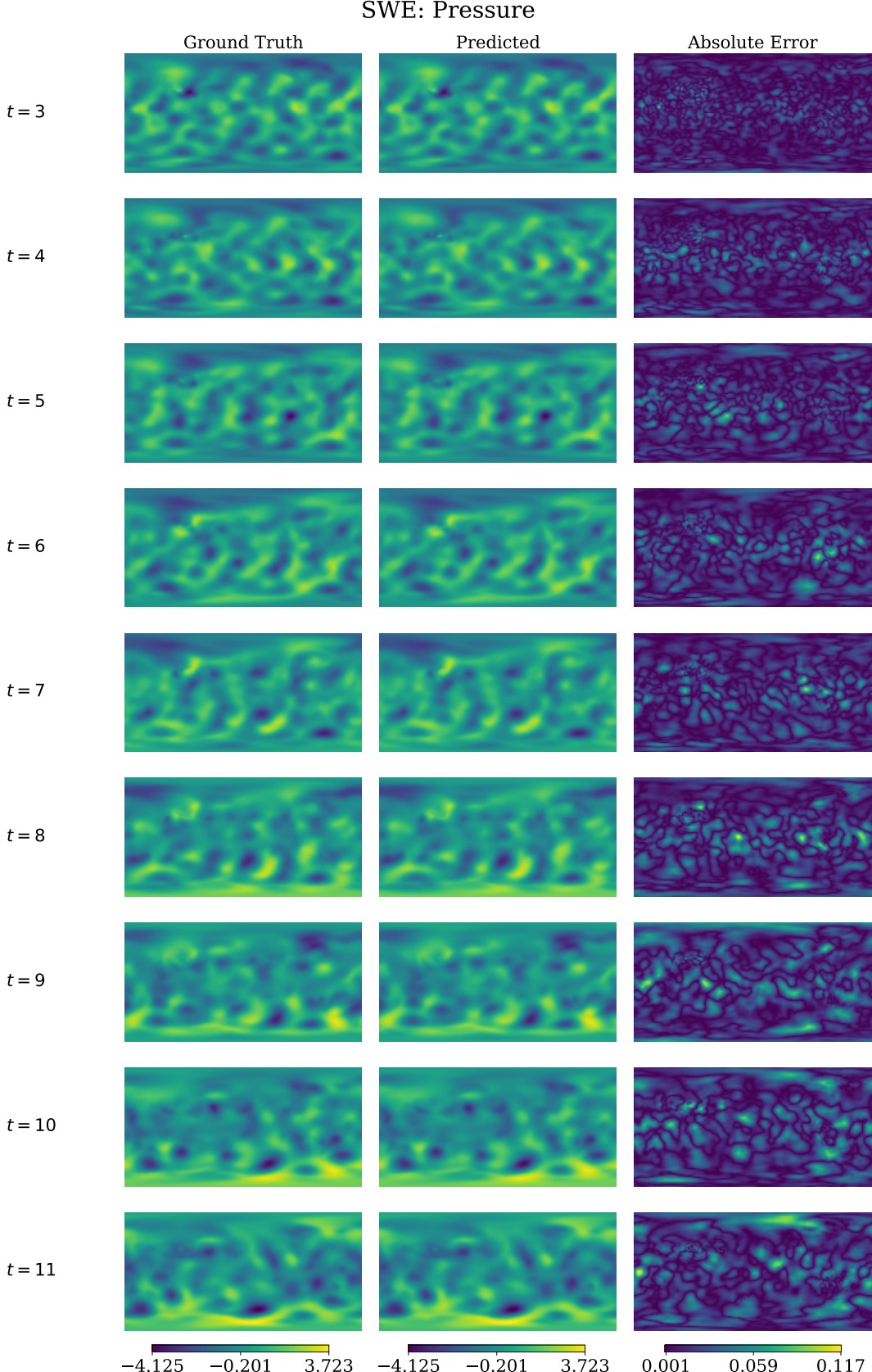

Figure 25: Shallow water equations pressure field.

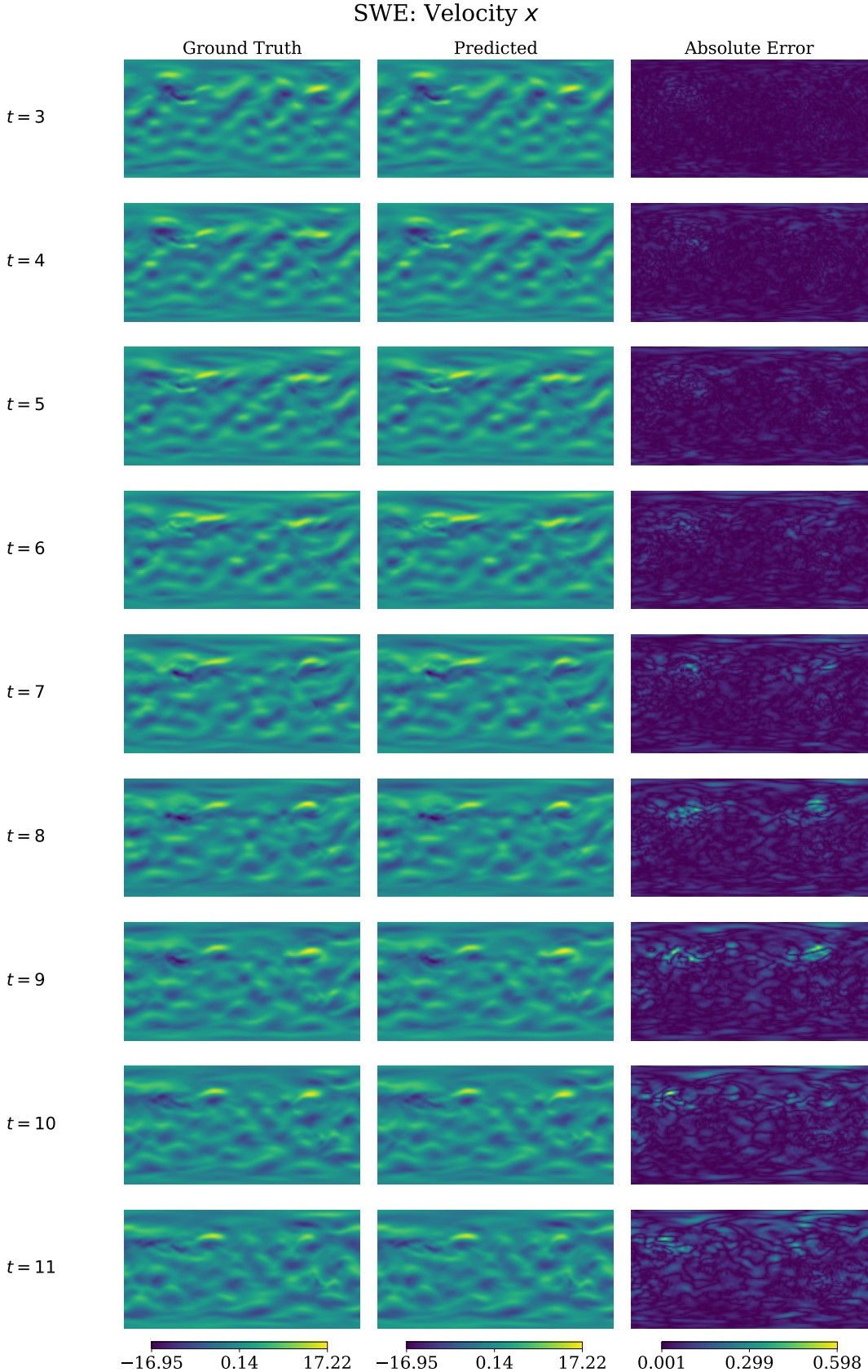

Figure 26: Shallow water equations velocity $x$ component.

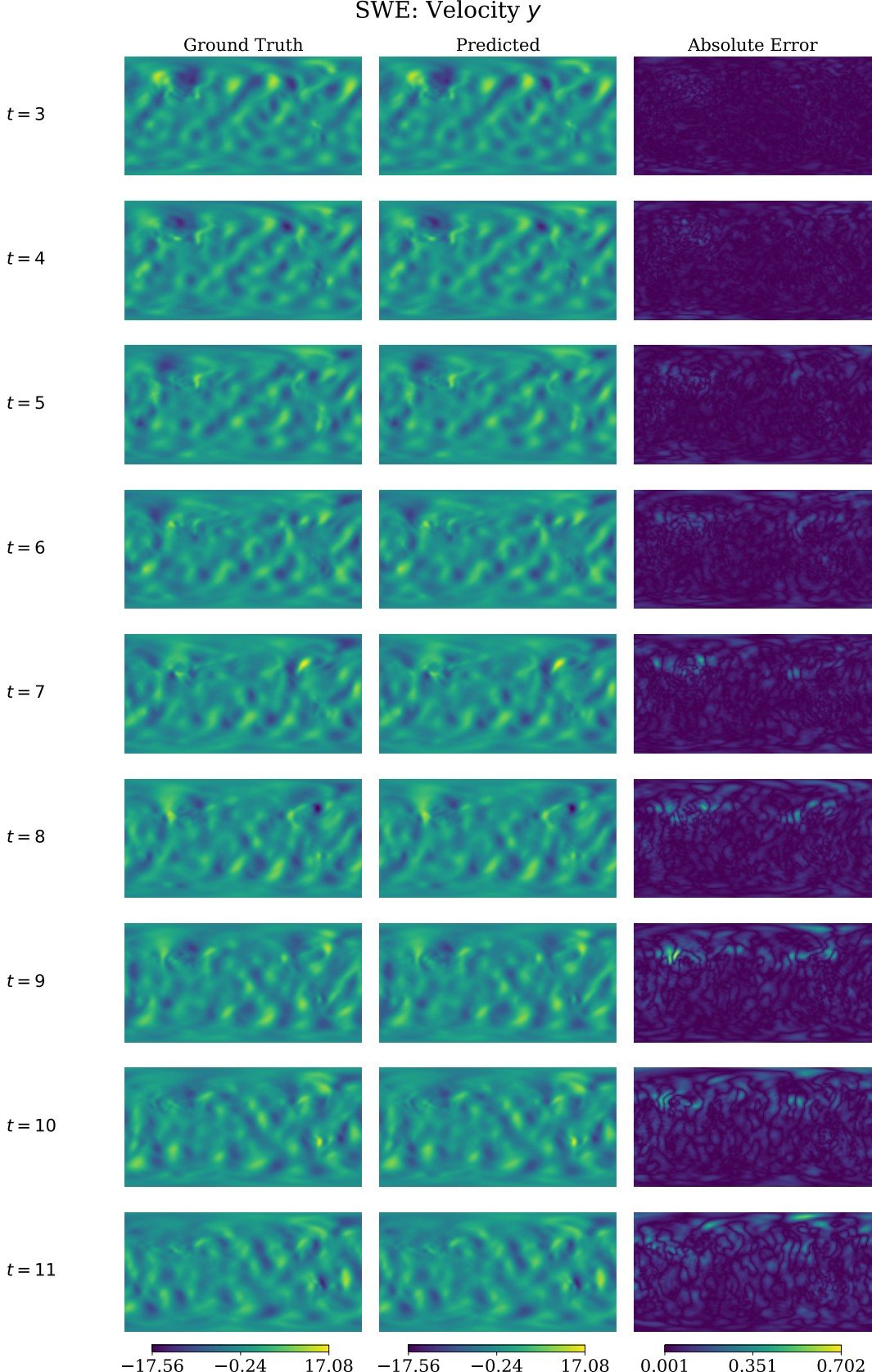

Figure 27: Shallow water equations velocity $y$ component.

