# OpenReview forum: "SineNet: Learning Temporal Dynamics in Time-Dependent Partial Differential Equations"
_ICLR.cc/2024/Conference — ICLR 2024 poster_

### Official Review · Reviewer_BgSq · 2023-10-30

**Soundness:** 3 good
**Presentation:** 3 good
**Contribution:** 3 good
**Rating:** 8
**Confidence:** 4

**Summary:**

This paper introduces SineNet as sequential / multi-stage UNet which improves temporal modeling for time-dependent partial differential equation (PDE) surrogates. This is motivated by misalignment in UNet architectures between downsampling and upsampling paths. The misalignment is caused by the temporal update of the underlying PDE which is in stark contrast to segmentation tasks for which the U-Net was initially introduced. SineNet architectures consist of multiple waves which are individual small UNet components, and thus, enable a smooth transfer between input timesteps and predicted timestep. SineNet models are testes against Factorized Fourier Neural Operators, Dilated ResNets, and various U-Net variants on 2D benchmarks of compressible / incompressible Navier-Stokes equations and shallow water equations.

**Strengths:**

- The paper is well motivated. It presents an intriguing observation on temporal PDE modeling for U-Nets. Whereas U-Nets are ideally suited for e.g. segmentation tasks (U-Nets extract information on various scales hierarchically before invoking a symmetric upsampling path), for PDE modeling this symmetric upsampling is broken since the time evolution of the system is incorrectly modeled. The visualizations help a lot understanding this problematic.
- The architecture design is novel and thought-through.
- Experimental results are strong and ablations are convincing, personally I like Figure 3 a lot!
- The code is clean and easy to follow.

**Weaknesses:**

- As clearly stated in the paper, U-Nets in this setting only work for fixed grid resolutions. There are however already works which design operators for different grids - see e.g.,  Raonic et al. Would it be possible to extend the experiments to some "super-resolution" experiments (i.e., evaluating at a different (higher) resolution than the one used during training)? I am pretty convinced SineNets are doing great there as well.
- One advantage U-Nets have is that they are storage efficient since the highest resolution is only used in the first and last block. I am not sure how SineNets handle this - probably through P and Q networks? It would be great to have some information on this. Toward this end I am wondering if the number of waves is increasing the memory footprint and by how much?



Raonić, B., Molinaro, R., Rohner, T., Mishra, S., & de Bezenac, E. (2023). Convolutional Neural Operators. arXiv preprint arXiv:2302.01178.

**Questions:**

- The iterative refinement process is conceptually similar to PDE Refiner (Lippe et al.) - with some major differences to my understanding. SineNets iteratively refines the temporal update in an one end-to-end approach whereas PDE Refiner looks at the input several times and refines different frequency components. It might be interesting to contrast those two approaches in the discussion.
- The paper strongly references the original paper of Gupta & Brandstetter. This paper put some emphasis on conditioning on equation parameters as well as flexible lead time predictions. It would be interesting to have statements / experiments on this as well. Similar FiLM like encodings would work for SineNet I suppose?
- The misalignment made me realize why ViTs or Swin Transformers might be better fit for PDE modeling compared to standard U-Nets. Can you comment on this?

I am happy to raise my score if weaknesses and questions are addressed properly.




Lippe, P., Veeling, B. S., Perdikaris, P., Turner, R. E., & Brandstetter, J. (2023). Pde-refiner: Achieving accurate long rollouts with neural pde solvers. arXiv preprint arXiv:2308.05732.

---

> ### Author Response · Authors · 2023-11-19
> **Rebuttal to Reviewer BgSq (1/3)**
>
> Thank you for your constructive comments and insightful questions. We address each of your points below.
>
> ## **S1. Figure 3**
>
> > personally I like Figure 3 a lot!
>
> Thank you for the positive feedback -- per the suggestion of Reviewer **NgLk**, we have added SineNet-10, SineNet-12, SineNet-14, and SineNet-16 to this figure.
>
> ## **W1. Super-resolution**
>
> > As clearly stated in the paper, U-Nets in this setting only work for fixed grid resolutions. There are however already works which design operators for different grids - see e.g., Raonic et al. Would it be possible to extend the experiments to some "super-resolution" experiments (i.e., evaluating at a different (higher) resolution than the one used during training)? I am pretty convinced SineNets are doing great there as well.
>
> Thanks for the suggestion! Indeed we can show that SineNet can work for super-resolution without any re-training.
>
> To perform super-resolution with SineNet, we consider two approaches: **interpolation** and **dilation**.
>
> - **Interpolation:** The input initial 10 time steps are downsampled to the training resolution of 128$\times$128. SineNet then solves the PDE at the lower resolution, after which the solution is interpolated to the higher resolution.
>
> - **Dilation:** Alternatively, SineNet can operate directly on the high resolution data using dilation. Convolutions, downsampling, and upsampling operations from the SineNet trained on 128$\times$128 are all dilated by a factor of 4.
> Intuitively, the grid can be viewed to be divided using a checkerboard pattern, where each grid point interacts exclusively with other grid points that share the same type (color) as designated by this checkerboard arrangement.
> This ensures that even at a higher resolution, each feature map grid point interacts only with grid points spaced equidistant to those it would interact with at the training resolution.
> Although dilation is a standard operation in convolution layers, we highlight that it is crucial to also apply dilation in pooling (for downsampling) and in interpolation (for upsampling).
> For example, in the 2$\times$2 pooling operation, instead of averaging and reducing over 2$\times$2 regions of immediate neighboring grid points, we average and reduce over the 4 corner grid points of 5$\times$5 regions.
>
> We download 1,000 512$\times$512 CNS trajectories from PDEBench [1] and evaluate models trained on the 128$\times$128 CNS data presented in the main text.
>
> We compare to F-FNO, a neural operator which does not require interpolation or dilation to perform super-resolution. The results are summarized in Table 9 in Appendix.
>
> From the results we see that at the higher resolution, interpolation introduces error, while dilation achieves nearly identical error to at training resolution. In Figure 8 of Appendix, we visualize the super-resolved errors using both approaches for one-step prediction on a randomly chosen example. As can be seen, unlike dilation, interpolation introduces errors in regions of high gradient. However, dilation is only applicable for super-resolving at integer multiples of the training resolution.
> For non-integer super-resolution multiples, similar techniques in deformable convolutions [2] could be considered to offset the operations to non-integer locations during testing.
>
> CNO [3] achieves super-resolution by limiting the frequency band of feature maps so that a fixed grid would be sufficient to preserve the information (by the Shannon sampling theorem). In a certain sense, their method shares similar ideas with the interpolation method, but with more advanced techniques (*e.g.,* convolving with interpolation sinc filters for downsampling [3]).
>
> We include the above experiments and more discussions in Appendix F.3 which are referred to in the Discussion section added in the main paper.

---

> ### Author Response · Authors · 2023-11-19
> **Rebuttal to Reviewer BgSq (2/3)**
>
> ## **W2. Storage efficient and memory footprint**
>
> > One advantage U-Nets have is that they are storage efficient since the highest resolution is only used in the first and last block. I am not sure how SineNets handle this - probably through P and Q networks? It would be great to have some information on this. Toward this end I am wondering if the number of waves is increasing the memory footprint and by how much?
>
> We summarize time and memory usage in Table 4, Appendix D, where we have additionally added training statistics. We can see that although SineNet uses more high-resolution feature maps than a standard U-Net, the forward memory will not increase much during inference, as despite the deep architecutre, activations do not need to be stored. Additionally, we can adjust the channel multiplier (detailed in Table 5, Appendix D.3) to control the number of channels in the down and up blocks. As a result, although the training memory increases more with number of waves compared to inference since more feature activations need to be stored for backpropagation, we are able to constrain the training memory to grow less than 5GB per additional wave with a batch size of 32.
>
> ## **Q1. Relation to PDE Refiner**
>
> > The iterative refinement process is conceptually similar to PDE Refiner (Lippe et al.) - with some major differences to my understanding. SineNets iteratively refines the temporal update in an one end-to-end approach whereas PDE Refiner looks at the input several times and refines different frequency components. It might be interesting to contrast those two approaches in the discussion.
>
> Thank you for bringing this work to our attention -- we agree that this connection is interesting, and thus have compared and contrasted the two approaches in our updated related work section.
>
> ## **Q2. Conditioning on equation parameters and flexible lead time predictions**
>
> > The paper strongly references the original paper of Gupta & Brandstetter. This paper put some emphasis on conditioning on equation parameters as well as flexible lead time predictions. It would be interesting to have statements / experiments on this as well. Similar FiLM like encodings would work for SineNet I suppose?
>
> Thank you for the suggestion. We have added results on the conditional task in Appendix F.1. We agree that FiLM encodings are applicable to SineNet, although we only considered injecting the embedding of parameters/lead times to feature maps via addition along the channel dimension, and not with the Hadamard product. Interestingly, we observe that the margin between SineNet and the baseline U-Net-Mod is larger for longer lead times, where there is a greater degree of misalignment between the input and target.

---

> > ### Author Response · Authors · 2023-11-19
> > **Rebuttal to Reviewer BgSq (3/3)**
> >
> > ## **Q3. Discussion on ViTs & Swin Transformers**
> >
> > > The misalignment made me realize why ViTs or Swin Transformers might be better fit for PDE modeling compared to standard U-Nets. Can you comment on this?
> >
> > Indeed, while both U-Nets and attention-based architectures include multi-scale processing mechansisms, allowing them to achieve a global receptive field, Transformers do not include such long range skip-connections and therefore do not encounter the misalignment issue at all. However, we suggest that the advantage of using a U-Net for PDE simulation could lie in sample complexity. From the paragraph starting at the bottom of page 1 of [4]:
> >
> > > When trained on mid-sized datasets such as ImageNet without strong regularization, [ViTs] yield modest accuracies of a few percentage points below ResNets of comparable size. This seemingly discouraging outcome may be expected: Transformers lack some of the inductive biases inherent to CNNs, such as translation equivariance and locality, and therefore do not generalize well
> > when trained on insufficient amounts of data...However, the picture changes if the models are trained on larger datasets (14M-300M images).
> >
> > Although further experimentation is needed to determine whether this trend holds for PDE modeling, if it does, it could explain why U-Nets are preferrable for PDE modeling -- it is generally infeasible to obtain such large quantities of dynamics data due to the cost of simulation. One prominent exception has been climate modeling and weather forecasting, where vision transformers have been commonly applied [5,6,7]. The dataset used in each of these works, ERA5 [8], contains hourly measurements dating back 40 years, and is therefore several orders of magnitude larger than those considered in the majority of the PDE modeling literature.
> >
> > Swin Transformer [9] improves the computational efficiency of ViT by partitioning feature maps into smaller local windows and only computing self-attention inside each window. To model long-range dependencies, the window size is increased deeper into the network. However, since interactions between windows are not modeled, Swin achieves a global receptive field only in the final layers. Although this is not a limiting factor for image tasks since patterns are mainly localized in space, for dynamics modeling, there are stronger long-range dependencies due to processes such as diffusion. As a result, we may need to have global information as early as possible so that local processing can have sufficient global context. To mitigate this issue, we could mirror the Swin Transformer to construct a U-Shaped Swin Transformer that achieves a global receptive field in the middle of the network. Moreover, while limiting interactions to be modeled inside each local window is key for improving efficiency, communication between neighboring windows may be needed to account for advection and diffusion processes in dynamics modeling. Consequently, further modifications to the Swin architectures may be needed to adapt it for dynamics modeling.
> >
> >
> > [1] Takamoto, Makoto, et al. "PDEBench: An extensive benchmark for scientific machine learning." Advances in Neural Information Processing Systems 35 (2022): 1596-1611.
> >
> > [2] Dai, Jifeng, et al. "Deformable convolutional networks." Proceedings of the IEEE international conference on computer vision. 2017.
> >
> > [3] Raonic, Bogdan, et al. "Convolutional Neural Operators for Robust and Accurate Learning of PDEs." Thirty-seventh Conference on Neural Information Processing Systems. 2023.
> >
> > [4] Dosovitskiy, Alexey, et al. "An Image is Worth 16x16 Words: Transformers for Image Recognition at Scale." International Conference on Learning Representations. 2020.
> >
> > [5] Pathak, Jaideep, et al. "Fourcastnet: A global data-driven high-resolution weather model using adaptive fourier neural operators." arXiv preprint arXiv:2202.11214 (2022).
> >
> > [6] Nguyen, Tung, et al. "ClimaX: A foundation model for weather and climate." arXiv preprint arXiv:2301.10343 (2023).
> >
> > [7] Bi, Kaifeng, et al. "Pangu-weather: A 3d high-resolution model for fast and accurate global weather forecast." arXiv preprint arXiv:2211.02556 (2022).
> >
> > [8] Hersbach, Hans, et al. "The ERA5 global reanalysis." Quarterly Journal of the Royal Meteorological Society 146.730 (2020): 1999-2049.
> >
> > [9] Liu, Ze, et al. "Swin transformer: Hierarchical vision transformer using shifted windows." Proceedings of the IEEE/CVF international conference on computer vision. 2021.
> >
> > Please let us know if any further clarifications are needed.

---

> > > ### Comment · Reviewer_BgSq · 2023-11-21
> > > **Post-rebuttal**
> > >
> > > Thank you for the thorough rebuttal. My comments and worries have been addressed extensively. I have thus raised my score, and consider this paper as a valuable finding for the community.

---

> ### Author Response · Authors · 2023-11-21
> **Thank you!**
>
> Thank you for reading our rebuttal and raising the score! We are glad to hear that we were able to address your concerns. We greatly appreciate your insightful feedback and the role it played in improving our manuscript.

---

### Official Review · Reviewer_NgLk · 2023-10-30

**Soundness:** 2 fair
**Presentation:** 4 excellent
**Contribution:** 3 good
**Rating:** 6
**Confidence:** 3

**Summary:**

When applying U-Net architectures with skip connections to dynamical problems, the features do not evolve across layers, thus limiting the efficiency of these methods for predicting dynamical systems. The method proposed in the article tries to overcome this challenge by having features across layers that evolve with the time. More precisely, the architecture is an encode-process-decode architecture consisting of several U-Net architectures stacked together with convolutions to encode and decode the solutions. Each U-Net, or wave, predicts the latent solution at a small time step instead of at the final time step, thus allowing evolving features. In addition to this global architecture, the downsampling and upsampling operations include residual blocks from the input, thus connecting the evolution of the features directly with the input. The architecture is tested on fluid mechanics problems, namely incompressible Navier-Stokes, compressible Navier-Stokes and Shallow Water equations. The experiments consist in forecasting these dynamical systems.

**Strengths:**

- The method solves a common problems when directly applying U-Net architectures to dynamical systems.

- The paper is well written and the justifications of the architecture are documented.

- The performances of the method seem to be competitive with several baselines.

**Weaknesses:**

- Stacked U-net have already been developed  for different tasks as detailed in the article (section Stacked U-Nets in the related work). The implementation here differs in some minor points and thus is more effective for PDE prediction but this is a marginal improvement over the previous works.
- The experiments are not extensive. First, more baselines should be used, with at least DeepONet [1], MP-PDE ([2], cited but not implemented) or Neural ODE [3], which closely resembles the formulation of SineNet. Second, the evolution of the error with the time is not shown, only at a given number of rollout steps for each datasets, thus limiting the impact of the results. Third, no ablation on the influence of the number of historical time steps is performed.
- As other U-Net architectures, SineNet cannot handle irregular grids or grids of different sizes, which are very common in various dynamics resolution problems. This limits the applicability of this line of methods to more practical problems.
- This is more minor. As with other auto-regressive architectures, the forecasting process is done at regular steps, so the architecture cannot produce values at any time $t$, but only at multiples of $\Delta t$.

[1]:Lu, L., Jin, P.,  Karniadakis, G. E. (2019). Deeponet: Learning nonlinear operators for identifying differential equations based on the universal approximation theorem of operators. arXiv preprint arXiv:1910.03193.

[2]:Brandstetter, J., Worrall, D.,  Welling, M. (2022). Message passing neural PDE solvers. arXiv preprint arXiv:2202.03376.

[3]: Chen, R. T., Rubanova, Y., Bettencourt, J., Duvenaud, D. K. (2018). Neural ordinary differential equations. Advances in neural information processing systems, 31.

**Questions:**

- Have you tried with a high number of rollout steps ? It would be interesting to see how stable the method is when pushed to its limits.
- In Figure 3, the error seems to keep decreasing with the number of waves. Have you tested with more waves? This would be interesting to see when the error arrives at a plateau and this could help improve the performances of the model if the error keeps decreasing with the number of waves.
- Have you tried using noise to improve the rollout stability? (as has been done for instance in [4])
- In the experiments, null Dirichlet and Neumann boundary conditions were chosen. What happens if different values for these conditions are chosen? Will zero padding still be effective?

[4]: Stachenfeld, K., Fielding, D. B., Kochkov, D., Cranmer, M., Pfaff, T., Godwin, J., ..., Sanchez-Gonzalez, A. (2021). Learned coarse models for efficient turbulence simulation. arXiv preprint arXiv:2112.15275.

---

> ### Author Response · Authors · 2023-11-19
> **Rebuttal to Reviewer NgLk (1/3)**
>
> Thank you for your constructive comments and insightful questions. We address each of your points below.
>
> ## **W1. Clarification of novelty**
>
> > Stacked U-net have already been developed for different tasks as detailed in the article (section Stacked U-Nets in the related work). The implementation here differs in some minor points and thus is more effective for PDE prediction but this is a marginal improvement over the previous works.
>
> One of our primary contributions is the identification of the misalignment issue encountered by U-Nets when faced with temporal dynamics. As many works have considered U-shaped architectures for dynamics forecasting due to their strengths in multi-scale processing, e.g. [1,2,3], this is a far-reaching issue. Although our proposed solution of a stacked U-Net is not by itself a novel methodology, it improves upon state-of-the-art neural solvers by a substantial amount on 3 challenging datasets. Furthermore, through identification of the issue, our contribution has opened the door for future works on adapting U-Nets in other ways beyond stacking to address the misalignment issue.
>
> ## **W2-1. More baselines (DeepONet [4], MP-PDE [5], Neural ODE [6])**
>
> > First, more baselines should be used, with at least DeepONet, MP-PDE (cited but not implemented) or Neural ODE, which closely resembles the formulation of SineNet.
>
> ### Neural ODE
>
> Thank you for pointing out the connection between SineNet and Neural ODE. We state this in Section 3.3 of our revisions and present experiments and an in-depth discussion in Appendix F.2. We find that adapting SineNet into the neural ODE framework gives results which are competive with SineNet-8 on INS. However, due to the high number of function evaluations per training example, SineNet-Neural-ODE is expensive to train and inference, which we quantify in Table 4. The [code for this experiment](https://anonymous.4open.science/r/SineNet-03B2/pdearena/pdearena/modules/sinenet_neural_ode.py) can be found on our anonymous GitHub.
>
> ### DeepONet and MP-PDE
>
> We implemented a DeepONet with a CNN branch net using strided convolution downsampling based on code ([from the official GitHub](https://github.com/lu-group/deeponet-fno/blob/main/src/darcy_rectangular_pwc/deeponet.py) and [the DeepXDE package](https://deepxde.readthedocs.io/en/latest/_modules/deepxde/nn/pytorch/deeponet.html#DeepONet)). Hyperparameter choices (depth 4 for the trunk net, 512 basis functions, constant hidden width of 512) were based on Table C.1 of [7]. We present results for this model on INS below.
>
> Our MP-PDE implementation used code from [from the official GitHub](https://github.com/brandstetter-johannes/MP-Neural-PDE-Solvers/blob/master/experiments/models_gnn.py) adapted to 2D graphs. As the PDEs we considered were discretized on a 128$\times$128 grid, the resulting graph was very large, with >16K nodes and >145K edges. Therefore, we did not modify the default hyperparameters used in the official repo (e.g., depth and width), which gave 638K parameters. Due to the large graph, even this small model did not fit on an 80GB A100 GPU with batch size 32, and we therefore reduced the batch size to 16. We also found injecting noise at level $\sigma=0.01$ to improve performance for this model.
>
> Results for both models are presented below on INS. As can be seen, further work is needed to achieve better results on this dataset for these models. For DeepONet, we hypothesize that because the model does not exploit the regular grid on which the target function is discretized, and instead can evaluate the target function on any point in its domain, its capacity for modeling regularly-grided functions is naturally less than that of a CNN-type architecture which exploits this structure. Similarly, MP-PDE has the flexibility to model mappings between irregularly-meshed functions, which we discuss as a limitation of SineNet and other CNN-based architectures in Appendix E. However, in the context of modeling on a square regular grid, this flexibility may be a limiting factor. Furthermore, MP-PDE does not include mechanisms for modeling long-range dependencies, thereby limiting the receptive field. This could also explain the performance, and is a limitation addressed in follow-up work [8].
>
> Our [implementation of MP-PDE](https://anonymous.4open.science/r/SineNet-03B2/pdearena/pdearena/modules/mp_pde.py) can be found on our anonymous GitHub, as can [our implementation of DeepONet](https://anonymous.4open.science/r/SineNet-03B2/pdearena/pdearena/modules/deepONet.py).
>
> | Model | \# Par. (M) | 1-Step (\%) | Rollout (\%) |
> |---|---|---|---|
> | SineNet-8 | 35.5 | **1.66**| **19.25** |
> | DeepONet | 38.5 | >20.00 | >40.00 |
> | MP-PDE | 0.6 | >20.00 | >40.00 |

---

> ### Author Response · Authors · 2023-11-19
> **Rebuttal to Reviewer NgLk (2/3)**
>
> ## **W2-2. Error Evolution with time.**
>
> > Second, the evolution of the error with the time is not shown, only at a given number of rollout steps for each datasets, thus limiting the impact of the results.
>
> Thank you for the suggestion. We have included plots in Appendix F.4 (Figure 9) to show the error evolution with time on all three datasets.
>
> ## **W2-3. Influence of the number of historical time steps**
>
> > Third, no ablation on the influence of the number of historical time steps is performed.
>
> Thank you for the suggestion. We have trained new models on the incompressible Navier Stokes (INS) dataset conditioning on 1, 2 or 3 historical step(s), and present results in Appendix F.6. We found that our choice in the main text of conditioning on 4 historical steps was not optimal for SineNet. However, it is not clear whether this trend would hold for other baselines, and thus, we choose to adhere to the number of conditioning steps used by the respective benchmarks on each dataset (4 for INS, 10 for CNS and 2 for SWE).
>
> ## **W3. Applicability on irregular grids or grids of different sizes**
>
> > As other U-Net architectures, SineNet cannot handle irregular grids or grids of different sizes, which are very common in various dynamics resolution problems. This limits the applicability of this line of methods to more practical problems.
>
> ### Irregular grid
>
> Indeed, SineNet and other CNN-based architectures can only handle inputs on rectangular grids with uniformly spaced mesh points. We have stated this limitation in our revised Section 6, and have added a more in-depth discussion to Appendix E.
>
> ### Grid of different sizes
>
> It is true that neural solvers tend to struggle in performing inference on domains of sizes differing from those observed during training, which we have discussed in Appendix E. When the domain size is fixed, but the resolution increases relative to the resolution observed during training, this becomes the *super-resolution task*, as considered in the neural operator line of work [9]. We have added an in-depth discussion and experiments on this to Appendix F.3.
>
> Although SineNet achieves the top performance on all datasets at training resolution, as a CNN, it inherits a dependence on the resolution of the training data, and therefore cannot perform super-resolution directly. While predictions can be interpolated, this introduces errors, particularly in regions of high gradient. However, by dilating all network operations (upsampling, downsampling, and convolutions), we demostrate the ability to perform inference on the CNS data at a resolution 4$\times$ greater than the training resolution without re-training or compromising performance. The code for [our super-resolution SineNet](https://anonymous.4open.science/r/SineNet-03B2/pdearena/pdearena/modules/sinenet_SR.py) is available on our Anonymous GitHub.
>
> ## **W4. Irregular time steps**
>
> > This is more minor. As with other auto-regressive architectures, the forecasting process is done at regular steps, so the architecture cannot produce values at any time $t$, but only at multiples of $\Delta t$.
>
>
> In the conditional task considered by [1], models are trained to generalize over time step size and forcing terms by conditioning the model on a time step size and the forcing term. Although in this task, models are only supervised on time step sizes which are multiples of 0.375 seconds (the temporal resolution of the data), it is possible to condition the model to advance time by any time step size. However, since this would be out-of-distribution, further data generation and experimentation is needed to assess model performance in such a setting. Nontheless, we have added results on this conditional task to Appendix F.1.

---

> ### Author Response · Authors · 2023-11-19
> **Rebuttal to Reviewer NgLk (3/3)**
>
> ## **Q1 and Q3. Higher number of rollout steps and noise injection**
>
> > Have you tried with a high number of rollout steps? It would be interesting to see how stable the method is when pushed to its limits.
>
> Thank you for the suggestion. We have generated 100 trajectories of CNS data which are 10$\times$ longer than those in the main text. We found that although noise injection reduces performance for shorter rollouts, it plays a key factor in maintaining long-term stability. We discuss this in greater depth and present results in Appendix F.5, including the averaged rollout error, as well as a plot showing the evolution of the rollout error across time.
>
> ## **Q2. More waves**
>
> > In Figure 3, the error seems to keep decreasing with the number of waves. Have you tested with more waves?
>
> Thank you for the suggestion. On all three datasets, we have added results for SineNet-10,12,14, and 16. Results are included in Figure 3, and have also been reported numerically in Appendix F.8.
>
> ## **Q4. Zero padding on Dirichlet and Neumann boundary conditions with different values**
>
> > In the experiments, null Dirichlet and Neumann boundary conditions were chosen. What happens if different values for these conditions are chosen? Will zero padding still be effective?
>
> This is a good point. We have added a discussion on challenges presented by mixed boundary conditions and potential approaches for dealing with non-null boundary conditions in Appendix E on limitations. Although it would appear to be relatively simple to encode both via padding, we leave this to future work.
>
>
> [1] Gupta, Jayesh K., and Johannes Brandstetter. "Towards multi-spatiotemporal-scale generalized pde modeling." arXiv preprint arXiv:2209.15616 (2022).
>
> [2] Wang, Rui, et al. "Towards physics-informed deep learning for turbulent flow prediction." Proceedings of the 26th ACM SIGKDD International Conference on Knowledge Discovery & Data Mining. 2020.
>
> [3] Wen, Gege, et al. "U-FNO—An enhanced Fourier neural operator-based deep-learning model for multiphase flow." Advances in Water Resources 163 (2022): 104180.
>
> [4] Lu, Lu, Pengzhan Jin, and George Em Karniadakis. "Deeponet: Learning nonlinear operators for identifying differential equations based on the universal approximation theorem of operators." arXiv preprint arXiv:1910.03193 (2019).
>
> [5] Brandstetter, Johannes, Daniel Worrall, and Max Welling. "Message passing neural PDE solvers." arXiv preprint arXiv:2202.03376 (2022).
>
> [6] Chen, R. T., Rubanova, Y., Bettencourt, J., Duvenaud, D. K. (2018). Neural ordinary differential equations. Advances in neural information processing systems, 31.
>
> [7] Lu, Lu, et al. "A comprehensive and fair comparison of two neural operators (with practical extensions) based on fair data." Computer Methods in Applied Mechanics and Engineering 393 (2022): 114778.
>
> [8] Equer, Léonard, T. Konstantin Rusch, and Siddhartha Mishra. "Multi-scale message passing neural pde solvers." arXiv preprint arXiv:2302.03580 (2023).
>
> [9] Kovachki, Nikola, et al. "Neural operator: Learning maps between function spaces." arXiv preprint arXiv:2108.08481 (2021).
>
>
> Please let us know if any further clarifications are needed.

---

> ### Author Response · Authors · 2023-11-22
> **36 hours left in discussion period**
>
> Dear Reviewer NgLk,
>
> Thank you again for your review. In response to your points, we have included the following experiments and analyses in our rebuttal:
>
> - Comparison to DeepONet and MP-PDE
> - Experiment with Neural ODE framework
> - Plots of error evolution with time for all 3 datasets
> - Training and evaluation of SineNet with $h=$ 1,2,3 for an ablation on the number of historical time steps
> - Evaluation on a grid 4$\times$ finer than the training resolution
> - Evaluation on a dataset with 10$\times$ as many time steps as in our original results
> - Training and evaluation of SineNet with noise injection
> - Training and evaluation of SineNet-$K$, $K=$ 10,12,14,16, on all 3 datasets
>
> Finally, we would like to highlight that drawing the connection between the temporal evolution in PDEs and the misalignment in feature propagation (particularly in U-Nets) is a non-trivial observation, which we believe is an important step to help understand and design deep learning models that are more adapted for learning time-dependent PDEs. With the discussion period closing in roughly 36 hours, we would appreciate your feedback as to whether we have adequately addressed your concerns.
>
> Thank you!
>
> Authors

---

> > ### Comment · Reviewer_NgLk · 2023-11-22
> > **Thank you for your answers and experiments**
> >
> > Thank you for your answers and the many additional experiments, and sorry for the delayed response. In terms of experiments, I now find the work very complete and thorough, which helps in better understanding the value of the approach. Only studying the boundary conditions could be added, but I completely understand that given the time constraint, this was not possible to do, and the discussion on this topic is enough. The discussion on irregular grids/grids of different sizes is interesting; I particularly appreciate the super-resolution results with dilation. Overall, I am still not totally convinced of the originality of the approach; however, this work is well-written and valuable for the community by showing how to better apply U-net architectures to physical problems. Thus, I raise my score from 3 to 6.

---

> > > ### Author Response · Authors · 2023-11-23
> > > **Thank you!**
> > >
> > > Thank you for reading our rebuttal and for your willingness to re-evaluate our work! We are glad to hear our added experiments and analyses have made our work very complete and thorough. The additions you suggested strongly supported this completeness. We greatly appreciate your insightful feedback and the role it played in improving our manuscript.

---

### Official Review · Reviewer_XLvR · 2023-10-30

**Soundness:** 3 good
**Presentation:** 3 good
**Contribution:** 3 good
**Rating:** 6
**Confidence:** 3

**Summary:**

The paper addresses the misalignment problem caused by the U-Net architecture in the context of PDE simulations and proposes multi-stage modeling of temporal dynamics with SineNet. SineNet consists of multiple sequentially connected U-shaped network blocks, which the authors call waves. High-resolution features evolve progressively through multiple stages, and contains misalignment within each stage. The method is evaluated on typical problems where multi-scalability can be seen, and its effectiveness are shown compared to representative baselines.

**Strengths:**

The paper points out an interesting problem that happens when employing the U-Net architecture to simulate spatiotemporal dynamics. The paper is well-organized and is generally easy to follow. The idea of the proposed model is simple and the choice of the architecture is based on reasonable analyzation of existing works. Wide range of ablation study is conducted and the proposed method outperforms strong baselines in rollout experiments.

**Weaknesses:**

I still do not fully understand what the authors mean by misalignment, since the metric for misalignment does not look properly introduced. Although the loss $a_{t+\Delta l}$ is introduced in the Appendix A.1, the loss does not seem to serve as the metric because it (and the experiment in Appendix A.1) looks more like for the robustness against noise injected. Thus, it is also still unclear if the improvement in the rollout result reported in Table 1 and Table 2 is achieved by the reduction of misalignment.


Some ablation study is missing. For example, what about if we replace each $V_{k}$ with standard U-Net with the equations (2) and (3)? Can the authors also provide visualization on this ablation version of the model in a way similar to Figure 5?



**Minor comments**

Typo Section 3.3: … benefits of the U-Net architecture, we contruct the ….

**Questions:**

See the weakness above.

**Details Of Ethics Concerns:**

Nothing particular.

---

> ### Author Response · Authors · 2023-11-19
> **Rebuttal to Reviewer XLvR (1/2)**
>
> Thank you for your constructive comments and insightful questions. We address each of your points below.
>
> ## **W1. Clarification on the misalignment issue**
>
> > I still do not fully understand what the authors mean by misalignment, since the metric for misalignment does not look properly introduced. Although the loss $a_{t+\Delta l}$ is introduced in the Appendix A.1, the loss does not seem to serve as the metric because it (and the experiment in Appendix A.1) looks more like for the robustness against noise injected. Thus, it is also still unclear if the improvement in the rollout result reported in Table 1 and Table 2 is achieved by the reduction of misalignment.
>
> We would like to make clarifications for the misalignment issue as well as the experiment in Appendix A.1.
>
> ### Misalignment
>
> Here, misalignment refers to the **displacement of local patterns between two feature maps**, which happens between the PDE solutions at two consecutive time steps due to advection.
> To resolve this misalignment, feature maps undergo latent evolution. However, **due to the local
> processing of convolution layers, feature maps closer to the input (or output) will be more aligned
> with the input (or output), respectively.** This leads to misalignment between the feature maps
> propagated via skip connections and those obtained from the upsampling path.
>
> Thank you for bringing this potential ambiguity to our attention. We have added these clarifications in Section 3.2. To show the misalignment in U-Nets, we visualize the feature maps averaged over the channel dimension at each block within a well-trained U-Net in Figure 1. Although the average over channels may not fully describe the feature maps, it provides a simple representation of local patterns and thus can show the spatial evolution, as well as the misalignment between skip-connected feature maps.
>
> Based on this analysis, we hypothesized that:
>
> 1. By composing U-Nets, we could reduce the amount of time evolution handled by each U-Net.
> 2. Reducing time evolution handled by a given U-Net would in turn reduce the misalignment between the skip-connected features.
>
> We empirically validated (1) with the analysis presented in Appendix A.1, which showed that the amount of time evolution handled by each U-Net was reduced. We next validated (2) by examining the skip-connected feature maps in A.2, where we found that the degree of misalignment compared to that in Figure 1 to be reduced.
>
> ### Appendix A.1
>
> We would like to further clarify the experiment in Appendix A.1. Importantly, this is not a loss for misalignment, but rather an empirical demonstration of (1), that is, the reduction of the time interval each wave is tasked with managing.
>
> **We empirically demonstrate this in Appendix A.1** by examining how the addition of a small perturbation propagates through each wave. In Figure 4, we can observe that the amount of propagation of the noise achieved by each wave is far less than that for a single U-Net, which is in agreement with our hypothesis (1).
>
> Finally, we would like to clarify the motivation of testing with noise addition. In fact, instead of testing the robustness against noise injection, adding noise allows us to see the feature propagation from an excitation with a small spatial extent. Otherwise if we only provide the perturbation as input to the network, it will become an out-of-distribution problem for the network since there are no similar samples in the training set. However, if we inject a small amount of noise to a sample in the test set, the resulting input remains close to the original test sample, thus is still in-distribution and can be used to analyze the behavior of the trained model. We have included this clarification in Appendix A.1 to avoid potential confusion.

---

> ### Author Response · Authors · 2023-11-19
> **Rebuttal to Reviewer XLvR (2/2)**
>
> ## **W2. Replacing $V_k$ with standard U-Net**
>
> > Some ablation study is missing. For example, what about if we replace each
>  $V_k$ with standard U-Net with the equations (2) and (3)? Can the authors also provide visualization on this ablation version of the model in a way similar to Figure 5?
>
>
> There are two main difference between $V_k$ and the standard U-Net (equations (2) and (3)):
>
> 1. The dual multi-resolution processing which we ablate with SineNet-8-Entangled in Table 2.
> 2. We maintain a high-dimensional feature map between waves instead of decoding to the dimension of the solution space and encoding back to the latent space in the following wave.
>
> In Appendix F.7, we ablate (2) with SineNet-8-BottleNeck by decoding and re-encoding the latent solution between waves such that each $V_k$ becomes a standard U-Net. There is a loss of information by linearly projecting from high dimensions to low dimensions and then back again (the bottleneck). We therefore observe that the 1-step loss for SineNet-8-BottleNeck is slightly higher than than of SineNet-8. This has a regularizing effect on the rollout loss, where we observe a slight improvement in performance. Moreover, we visualize the decoded feature maps between each wave in Figure 15 of the revised draft.
>
> **Typo - Section 3.3.**
>
> We have fixed the typo in the revised paper ("contruct" -> "construct"). Thank you for pointing it out!
>
> Please let us know if any further clarifications are needed.

---

> > ### Author Response · Authors · 2023-11-22
> > **36 hours left in discussion period**
> >
> > Dear Reviewer XLvR,
> >
> > Thank you again for your review. In response to your points, we have included in our rebuttal an in-depth clarification on the misalignment issue, as well as the motivation of the experiment in Appendix A.1. Additionally, we have added an ablation on wave architecture with feature map visualizations as in Figure 5. With the discussion period closing in roughly 36 hours, we would appreciate your feedback as to whether we have adequately addressed your concerns.
> >
> > Thank you!
> >
> > Authors

---

> ### Comment · Reviewer_XLvR · 2023-11-22
> **Thank you very much for the answers.**
>
> I appreciate for the authors’ answers.
>
> While most of my concerns were addressed, the novelty of the paper is still unclear to me. As shown in Table 1 and Table 8 in Appendix F.7, the the rollout-error ratio for INS of stacking multiple standard U-Nets (= SineNet-8-Bottleneck) to U-Net-128 (which is roughly same architecture as original U-Net) is 19.19 to 24.94 while the ratio of Sine-Net-8 (resp. SineNet-8, h=1) to U-Net-128 is 19.25 (resp. 18.86) to 24.94, which look on the same level. To me, the proposed architecture looks mainly benefitting from stacking multiple (UNet-like) layers, which is a familiar way to reduce prediction errors by having shorter time step sizes. Do you have any observations or insights on effects, other than reducing error, particular to the proposed architecture?

---

> ### Author Response · Authors · 2023-11-23
> **Further clarifications (1/2)**
>
> Thank you for reading our rebuttal in detail and taking time to engage in the discussion. We would like to make further clarifications on the novelty.
>
> > As shown in Table 1 and Table 8 in Appendix F.7, the the rollout-error ratio for INS of stacking multiple standard U-Nets (= SineNet-8-Bottleneck) to U-Net-128 (which is roughly same architecture as original U-Net) is 19.19 to 24.94 while the ratio of Sine-Net-8 (resp. SineNet-8, h=1) to U-Net-128 is 19.25 (resp. 18.86) to 24.94, which look on the same level.
>
> That’s true - each wave in the SineNet-8-Bottleneck is similar to a standard U-Net (despite the additional dual multi-processing mechanism and using a reduced channel multiplier) and the close performance compared to SineNet-8 suggests that the primary improvements are a result of the multi-stage design, that is, composing multiple U-Net layers. We believe this is an interesting result and appreciate your idea to include it as an ablation.
>
> > To me, the proposed architecture looks mainly benefitting from stacking multiple (UNet-like) layers, which is a familiar way to reduce prediction errors by having shorter time step sizes.
>
> Indeed, as detailed in Section 3.3, the motivation for SineNet’s multi-stage architecture is to reduce the implicit time step in the latent space handled by each stage, which in turn simplifies the learning for each U-Net.
>
> It is well known that one can reduce finite-step error by reducing the $\Delta_t$ of the one-step mapping in classical solvers. This is because reducing the time step can help produce more accurate discrete estimates for continuous quantities such as time derivatives, increasing the physical consistency of solutions. This is related to our approach in SineNet, as in both cases reducing the (latent) time step helps describe the temporal evolution in a finer way.
>
> One important distinction between these two approaches is that while a classical solver is in some sense “supervised” by the laws of physics in every single step, the latent steps taken in each stage of SineNet are only supervised by the loss on the output of the final stage, back-propagated end-to-end to the other stages.
>
> Compared to reducing finite-step error in classical solvers, simplifying learning with finer time steps in latent space while holding the time step of the one-step mapping fixed is a natural but less-explored concept. The key observation we made in this paper was to show that the simplification in learning, resulting from the multi-stage design, is particularly beneficial for U-Nets due to misalignment in skip-connections.
>
> As detailed in Section 4, stacked U-Nets have been used for image tasks and steady-state PDE modeling (where temporal evolution converges to a static state). However, in current benchmarks for time-evolving PDEs, such as PDEArena [1] and PDEBench [2], as well as research on applying U-shaped architectures to time-evolving dynamics problems such as TF-Net [3] and UNO [4], researchers rely on single-layer U-Nets due to their strengths in modeling multi-scale dynamics despite their limitations due to the misalignment issue.
>
> In contrast, multi-stage architectures in which multiple global operators are composed have been used in non-U-Net-based PDE surrogate models, such as Fourier Neural Operators [5], which further supports our choice in adapting such a design to U-Nets.
>
> > Do you have any observations or insights on effects, other than reducing error, particular to the proposed architecture?
>
> The latent evolution and misalignment problem which we have brought to light in this work leads to a better interpretation of PDE surrogate models, opening the door for future works on adapting U-Nets in other ways beyond stacking to address the misalignment issue.

---

> > ### Author Response · Authors · 2023-11-23
> > **Further clarifications (2/2)**
> >
> > Finally, we would like to stress the importance of reducing the one-step prediction error in surrogate models. As shown in Appendix F.5 (in response to Reviewer **NgLk**), thanks to the good accuracy, the solutions produced by SineNet are still close to the ground truth after more than 100 rollout steps, which are visualized in Figure 11 - 14. Moreover, in response to Reviewers **BgSq** and **NgLk**, we show in Appendix F.3 that SineNet produces predictions at higher resolutions without re-training or compromising performance, which extends the usefulness of having a more accurate surrogate model.
> >
> > [1] Gupta, Jayesh K., and Johannes Brandstetter. "Towards multi-spatiotemporal-scale generalized pde modeling." arXiv preprint arXiv:2209.15616 (2022).
> >
> > [2] Takamoto, Makoto, et al. "PDEBench: An extensive benchmark for scientific machine learning." Advances in Neural Information Processing Systems 35 (2022): 1596-1611.
> >
> > [3] Wang, Rui, et al. "Towards physics-informed deep learning for turbulent flow prediction." Proceedings of the 26th ACM SIGKDD International Conference on Knowledge Discovery & Data Mining. 2020.
> >
> > [4] Rahman, Md Ashiqur, Zachary E. Ross, and Kamyar Azizzadenesheli. "U-no: U-shaped neural operators." arXiv preprint arXiv:2204.11127 (2022).
> >
> > [5] Li, Zongyi, et al. "Fourier Neural Operator for Parametric Partial Differential Equations." International Conference on Learning Representations. 2020.

---

> ### Author Response · Authors · 2023-11-23
> **SineNet-8-Bottleneck on SWE (new results)**
>
> To have a more comprehensive understanding of SineNet-8-Bottleneck, we additionally ran it on the shallow water equation (SWE) dataset. In our latest revision, we have included the results from this experiment in Table 11 and added a discussion in Appendix F.7. We also visualize the feature maps of SineNet-8-Bottleneck on SWE in Figure 16 in Appendix F.7.
>
> Contrary to the INS dataset, SineNet-8-Bottleneck showed a substantial performance drop compared to SineNet-8. We hypothesize that the different behavior in these two datasets is due to the fast-evolving nature at the initial time steps of the INS dataset, which makes the prediction more sensitive to small pertubations. While the bottleneck serves as a form of regularization which decreases performance in terms of 1-step prediction, it increases robustness to the difficult initial steps in INS. However for SWE, the difficulty of the initial timesteps instead appears similar to the remaining timesteps. Thus, the regularization is not beneficial as before, and in fact leads to a performance drop, likely due to the information bottleneck between waves.
>
> Interestingly, as SWE is a global weather forecasting task, we can see the outline of the world map in the visualized feature maps in Figure 16, which implies that the primary objective of these feature maps is for modeling the evolution of dynamics about the boundaries of continents, potentially encouraged by the information compression in the wave bottleneck.
>
> Given the new results on the SWE dataset, we believe that using a larger channel dimension between SineNet waves remains a more reasonable choice.

---

> ### Comment · Reviewer_XLvR · 2023-11-23
> **Thank you for the additional experiment and response**
>
> Thank you very much for the further clarification. Given the contributions of the paper as well as the authors’ willingness to provide explanations and a considerable amount of additional experimental results, I raised my score to 6 and will not object acceptance of this paper. As claimed by the authors, I also acknowledge simplifying learning with finer time steps in latent space while holding the time step of the one-step mapping fixed is less-explored and the paper posed a crucial problem when employing U-Net in fluid simulation. The proposed method is evaluated from various aspects. The additional experiment in Appendix F.7 addressed one of my main concerns and strengthened the validity of the proposed architecture. One additional question (which I couldn't ask due to the time limitation and has no influence on the score) for the proposed method would be to evaluate potential effect to reduction of training epochs, because the time step of one-step mapping can keep unchanged while latent mapping can have high-frequency waves. I will leave this question for a possible future direction.
>
> Thank you again for the additional work. I really enjoyed the discussion!

---

> > ### Author Response · Authors · 2023-11-23
> > **Thank you!**
> >
> > Thank you for engaging in discussion and for your willingness to re-evaluate our work! We are glad to hear our added experiments have strengthened the validity of our work. We greatly appreciate your insightful feedback and questions, and the role these have played in improving our manuscript.

---

### Official Review · Reviewer_uUSa · 2023-11-01

**Soundness:** 3 good
**Presentation:** 3 good
**Contribution:** 4 excellent
**Rating:** 6
**Confidence:** 5

**Summary:**

SineNet, a sequential U-Net with comparably small channel depths, is introduced to model the temporal evolution of 2D PDEs. By design, SineNet mitigates the spatial misalignment between skip connections in the traditional U-Net by subdividing the task of spatial propagation over multiple small U-Nets, referred to as waves. Experiments on multiple benchmarks demonstrate the superiority of SineNet against state-of-the-art methods.

**Strengths:**

_Originality:_ of spatial information misalignment between different hierarchies in the U-Net is an elemental observation and finding. Related work is exhaustive and cited adequately.

_Quality:_ All claims are well supported by appropriate visualizations and experiments.

_Clarity:_ The manuscript is mostly clear, well organized and written in approachable language. One point remained unclear to me and should be clarified upon publication (see question below).

_Significance:_ Solving the spatial misalignment problem with the proposed method has a potentially large impact on time-series forecasting with U-Nets and will advance the field. The ML community that is interested in simulating spatiotemporal dynamics (which also relates to video prediction) will likely benefit from the observations presented in this manuscript.

_Further comments_:
- The demonstration of spatially misaligned features in Figure 1 is very indicative and comprehensive.
- I like the neat property to have adaptive $\delta_k$ per wave to adequately model velocities of different speeds.
- Great to have runtime analysis and memory footprint of all models provided in Table 4 of Appendix C!

**Weaknesses:**

1. Unclear what limitations the method bears. Under what circumstances do you expect SineNet to break and fail?
2. Why does Dil-ResNet which you benchmark against only have 4.2M parameters (compared to 35M in your model)? This is a fairly critical point, since neither the inference time nor the required memory in Table 4 seem to justify such a small parameter count for Dil-ResNet. Indeed, I was curious to see how vanilla ResNet would perform and like that you have included Dil-ResNet. This actually forced me to only rate your submission as "marginally above threshold" instead of voting for a clear accept.

**Questions:**

1. In Section 3.4, does $L=4$ mean that four downsampling plus convolution combinations followed by four upsampling plus convolution combinations are employed? That is, $x=f_l(d(x))$ (encoding) followed by $x=g_l(v(x))$ (decoding) for $l\in[0, 1, 2, 3]$? Or do you employ only two downsampling and upsampling operations, as visualized in Figure 1? Maybe you can clarify right after Equation (7).
2. Have you tried to set $\delta_k$ as learnable parameter, or is the tuning of the time-subscale subject to the practitioner and model designer?

---

> ### Author Response · Authors · 2023-11-19
> **Rebuttal to Reviewer uUSa (1/2)**
>
> Thank you for your constructive comments and insightful questions. We address each of your points below.
>
>
> ## **W1. Discussion on limitations**
>
> > Unclear what limitations the method bears. Under what circumstances do you expect SineNet to break and fail?
>
> Thank you for raising this issue. We have stated several limitations in Section 6 which we discuss in greater detail in Appendix E in the revised draft. These limitations include out-of-distribution (OOD) generalization, applicability to PDEs with less temporal evolution, computational cost, and handling irregular spatiotemporal grids.
>
> Additionally, as pointed out by reviewers **BgSq** and **NgLk**, other potential limitations generalizing beyond the time step size and spatial grid resolution on which the model was trained. However, we have added conditional experiments in Appendix F.1, where we demonstrate the ability to generalize over time step size. Furthermore, in Appendix F.3, we demonstrate that by dilating all network operations (upsampling, downsampling, and convolutions), SineNet can perform inference on the CNS data at a resolution 4$\times$ greater than the training resolution without re-training or compromising performance.
>
> ## **W2. Dil-ResNet and ResNet**
>
> > Why does Dil-ResNet which you benchmark against only have 4.2M parameters (compared to 35M in your model)? This is a fairly critical point, since neither the inference time nor the required memory in Table 4 seem to justify such a small parameter count for Dil-ResNet. Indeed, I was curious to see how vanilla ResNet would perform and like that you have included Dil-ResNet.
>
>
> - **Dil-ResNet:** We use the DilResNet from the PDEArena benchmark [1] (DilResNet128-norm in their paper). Although it has fewer parameters, it is one of the strongest model in the PDEArena benchmark -- see Table 8 of [1].
> For a more fair comparison with SineNet, we trained a larger Dil-ResNet by increasing the hidden channels from 128 to 256 (dubbed Dil-ResNet-256), which has 16.7M parameters. Although it still has fewer parameters that SineNet, it is especially computationally expensive. We show this in Table 4 of Appendix C, where we report the computational cost of all models, which has been revised to include training stats. We can see that Dil-ResNet is slower in inference and is comparable with SineNet in training speed. However, it almost doubles the GPU memory usage of SineNets in both training and testing. This is because all convolutions and non-linearities take place on the full-resolution, as opposed to reducing the resolution as done in U-Nets and SineNet. The high computational cost prohibits us from using use more parameters for Dil-ResNet. Nontheless, the increase in channels substantially improves performance for Dil-ResNet, particularly on the INS data, where it is now extremely competive with SineNet-8, as can be seen in Table 1.
>
>
> - **ResNet:** According to results presented in PDEArena [1], ResNet underperforms other methods, including U-Nets, by more than one order of magnitude. Please see Table 8 in [1] for INS results and Table 2 for SWE results; note that their reported metric is MSE, as opposed to the scaled L2 loss as we use here, however, we expect similar results. The results are likely due to the limited receptive field in ResNets due to lack of downsampling or other multi-scale processing mechanisms such as dilation, which limit its ability to model long-range dependencies. Moreover, similar to Dil-ResNet, the computation is also much more expensive than U-Nets due to the lack of downsampling (see Table 1 in [1] for the comparison of computational costs).

---

> ### Author Response · Authors · 2023-11-19
> **Rebuttal to Reviewer uUSa (2/2)**
>
> ## **Q1. Number of down and up layers**
>
> > In Section 3.4, does $L=4$ mean that four downsampling plus convolution combinations followed by four upsampling plus convolution combinations are employed? That is, $x=f_l(d(x))$ (encoding) followed by $x=g_l(v(x))$ (decoding) for $l\in[1,2,3,4]$? Or do you employ only two downsampling and upsampling operations, as visualized in Figure 1? Maybe you can clarify right after Equation (7).
>
> Yes, your first understanding is accurate. We use 4 down blocks and 4 up blocks in each wave. We have updated with more clarifications in Section 3.4 under Equation (7). Thank you for pointing this out!
>
> ## **Q2. Setting $\delta_k$ as learnable parameter**
>
> > Have you tried to set $\delta_k$ as learnable parameter, or is the tuning of the time-subscale subject to the practitioner and model designer?
>
> This is an interesting question. Recall that $\delta_k$ is the amount of time that the $k$-th wave in SineNet advances the solution in the latent space. In training SineNet end-to-end to advance time from $u_t\to u_{t+1}$, $\delta_k$ is in fact learnable and likely varies between different examples, referred to as **adaptable temporal resolution**, which we further discuss in the second-to-last paragraph of Section 3.3.
>
> However, the $\delta_k$ are latent and therefore cannot be observed or even explicitly set without obtaining the ground truth solution at times $t+\Delta_t$ and applying deep supervision on each wave during training, where $\Delta_t\in(0,1)$ are chosen dependent on the desired $\delta_k$. Instead, one may parameterize a U-Net as the time derivative $\frac{dh}{d\tau}$ of $h(\tau)$, where $h_\tau$ is the latent of $u_{t+\tau}$. Then, to map from $u_t\to u_{t+1}$, one can integrate the derivative using a numerical integrator. Using this approach, the practitioner maintains the flexibility to determine the $\delta_k$ in selecting the time discretization for the numerical integration.
>
> Alternatively, one may choose to employ an adaptive time-stepping integrator, which selects the $\delta_k$ automatically so as to induce stability in the integration. This is the approach taken by neural ODE [2], and we have added an experiment using their framework to Appendix F.2. We find that although this approach leads to expensive training, it is able to achieve competitive results with SineNet-8. Moreover, as suggested by Reviewer **BgSq**, we also included experiments on the conditional INS task in Appendix F.1 where the models are trained to generalize over variable-sized time steps so that the user can control the prediction time interval, which implicitly modifies $\delta_k$.
>
> [1] Gupta, Jayesh K., and Johannes Brandstetter. "Towards multi-spatiotemporal-scale generalized pde modeling." arXiv preprint arXiv:2209.15616 (2022).
>
> [2] Chen, Ricky TQ, et al. "Neural ordinary differential equations." Advances in neural information processing systems 31 (2018).
>
> Please let us know if any further clarifications are needed.

---

> > ### Author Response · Authors · 2023-11-22
> > **36 hours left in discussion period**
> >
> > Dear Reviewer uUSa,
> >
> > Thank you again for your review. In response to your points, we have added a larger Dil-ResNet in our experiments, along with a more thorough evaluation of space and time complexity for all models. Additionally, we have added an in-depth discussion on the limitations of our proposed framework. With the discussion period closing in roughly 36 hours, we would appreciate your feedback as to whether we have adequately addressed your concerns.
> >
> > Thank you!
> >
> > Authors

---

### Author Response · Authors · 2023-11-19
**General Response**

We have uploaded a revised version of our paper with changes highlighted in blue. We have made the following changes in response to issues raised by reviewers:

 - Discussing potential limitation (uUSa)
 - More baselines:
    - Dil-ResNet-256 (uUSa)
    - Neural ODE (NgLk)
 - Super-resolution analysis (BgSq, NgLk)
 - SineNet with more waves (NgLk)
 - Extended analysis
    - Super-resolution analysis (BgSq, NgLk)
    - Conditioning experiments (BgSq)
    - Error evolution with time (NgLk)
    - Longer rollout (NgLk)
    - Number of conditioning steps (NgLk)
    - SineNet-BottleNeck (XLvR)
 - Training time and space complexity (BgSq)

---

### Meta-Review · Area_Chair_awgf · 2023-12-12

**Metareview:**

The manuscript proposes a neural network architecture for learning temporal dynamics of time dependent PDEs. The approach is numerically compared with a few existing approaches, which demonstrate its advantage. Overall the paper is well written and all reviewer concerns have been address in the discussion stage.

**Justification For Why Not Higher Score:**

While the proposed architecture is interesting, its novelty and significance does not warrant a higher score.

**Justification For Why Not Lower Score:**

see metareview

---

### Decision · Program_Chairs · 2024-01-16

Accept (poster)